# Marine ice-cliff instability modeling shows mixed-mode ice-cliff failure and yields calving rate parameterization

Anna J. Crawford [1✉], Douglas I. Benn[1], Joe Todd[2], Jan A. Åström[3], Jeremy N. Bassis [4] & Thomas Zwinger [3]

Marine ice-cliff instability could accelerate ice loss from Antarctica, and according to some model predictions could potentially contribute >1 m of global mean sea level rise by 2100 at current emission rates. Regions with over-deepening basins >1 km in depth (e.g., the West Antarctic Ice Sheet) are particularly susceptible to this instability, as retreat could expose increasingly tall cliffs that could exceed ice stability thresholds. Here, we use a suite of high-fidelity glacier models to improve understanding of the modes through which ice cliffs can structurally fail and derive a conservative ice-cliff failure retreat rate parameterization for ice-sheet models. Our results highlight the respective roles of viscous deformation, shear-band formation, and brittle-tensile failure within marine ice-cliff instability. Calving rates increase non-linearly with cliff height, but runaway ice-cliff retreat can be inhibited by viscous flow and back force from iceberg mélange.

[1] School of Geography and Sustainable Development, University of St Andrews, St Andrews, UK. [2] School of GeoSciences, The University of Edinburgh, Edinburgh, UK. [3] CSC – IT Center for Science, Espoo, Finland. [4] Department of Climate and Space Sciences and Engineering, University of Michigan, Ann Arbor, MI, USA. ✉email: ajc44@st-andrews.ac.uk

One prospect for a future Antarctica is the exposure of immense ice cliffs following the loss of buttressing ice shelves and continued grounding line retreat[1–3]. These calving faces may be susceptible to structural failure if glaciostatic stress exceeds the yield strength of ice (~500 kPa to 1 MPa), with this stress threshold being exceeded at a cliff height of ~100 m[2,4–8]. Marine ice-cliff instability (MICI) will initiate (i.e., ice-cliff retreat will be self-sustaining) if progressively taller cliffs are exposed through each cycle of failure[1,4,9]. Here, we make a distinction between *ice-cliff failure*, leading from stress imbalances at glacier termini, and *marine ice-cliff instability*, or the instability of a glacier or ice stream resulting from runaway ice-cliff failure. Determining if and how MICI will occur has recently challenged the glaciology community[10], with conclusions being difficult to draw since such cliff heights are outside of the modern observational record. The importance of clarifying whether MICI will materialise is demonstrated by ice-sheet model output showing enhanced and alarming rates of global mean sea-level rise (SLR) when ice-cliff failure is represented with a simple parameterization in models of the Antarctic Ice Sheet[1].

It is impractical to incorporate the granularity of the complex MICI process in continental-scale ice-sheet models, so reliable parameterizations of the key processes are needed to constrain SLR predictions[11–13]. Continuum models have recently been used to investigate processes and rates of MICI retreat, with cliff failure assumed to occur if strain rates or shear stress exceed threshold values[11,14]. However, the possibility that MICI could occur through a spectrum of failure modes means that reliable parameterizations must be based on modeling of these specific processes.

Employing three high-resolution 3D models and idealized glacier geometries, we conduct a series of simulations to investigate modes of structural ice-cliff failure. Our model suite consists of the full-Stokes continuum model Elmer/Ice[15,16], the standard brittle-elastic implementation of the Helsinki Discrete Element Model (HiDEM$_{be}$)[17,18], and a second, brittle visco-elastic implementation (HiDEM$_{ve}$). HiDEM represents glaciers as assemblages of densely-packed particles that are connected by beams that will break if strain, bending, or torsion, in combination, exceed a user-defined threshold. HiDEM$_{be}$ is used to simulate the predominantly tensile failure of glaciers, whereas HiDEM$_{ve}$ is designed to simultaneously model viscous deformation and fracture in a simplified form by partly decreasing or weakening the elastic-brittle bond connections while compensating with a short-range cohesive force that allows for viscous shear deformation. Ice properties are varied in HiDEM by adjusting beam width, fracture threshold and the degree of damage. The latter is represented as the percentage of initially broken bonds. More detailed descriptions of the HiDEM implementations are provided in the Methods and Supplementary Note 1. This work makes three contributions to understanding ice-cliff failure and the potential for MICI to impact future ice-sheet retreat. First, we identify multiple modes of failure and demonstrate how these emerge based on glacier characteristics and the selection of model parameters that influence shear and tensile strength. In general, when shear strength is high, calving cliffs fail through tensile failure following a period of viscous deformation. Due to the distinct timescales over which viscous and brittle processes occur, it is reasonable to simulate brittle and viscous processes separately using Elmer/Ice and HiDEM$_{be}$[19] as long as stresses are not high enough to initiate damage via shear-fracture. Shear failure is observed when the shear strength is reduced in HiDEM$_{be}$ or a thickness threshold is reached in HiDEM$_{ve}$. Our second contribution is the provision of back force values that mélange must exert on a calving face to halt structural failure. Lastly, we provide a retreat rate parameterization to represent ice-cliff failure in ice-sheet models. The parameterization was derived through a workflow that simulates the entirety of an ice-cliff failure event, which allows for the explicit determination of both calving magnitude and the duration of a calving cycle.

## Results and discussion

**Cliff failure via viscous deformation and tensile failure.** We simulate the deformation and failure of idealized glaciers grounded on a retrograde slope with varying ice temperature ($T_{ice}$) and bed friction conditions ($B$). The standard parameter settings and set-up of HiDEM$_{be}$ combine to represent ice that is strong in shear (influenced by the inter-particle yield strength, inter-particle beam width and lattice structure) and undamaged (influenced by the percentage of pre-seeded broken bonds, or 'porosity')[20] (see Methods).

For strong and undamaged ice, the failure of vertical calving faces is not observed in HiDEM$_{be}$ (Supplementary Fig. 3). This suggests that some degree of viscous deformation or shear localisation is a necessary pre-condition for the failure of ice cliffs with such characteristics over this topography. We therefore assess ice-cliff failure via interacting viscous deformation and brittle failure through a one-way offline coupling of Elmer/Ice and HiDEM$_{be}$[21,22]. Elmer/Ice is employed to simulate the viscous deformation of initially vertical ice faces, and output from successive time-steps is used to initialize the domain of HiDEM$_{be}$ simulations that test the evolved geometry for brittle failure. This workflow is re-iterated to determine the timing and magnitude of cliff failure for glaciers with increasing thickness ($H$) and a corresponding increase in cliff height ($H_c$). We discuss our results in terms of $H_c$, noting that references or values could be similarly described in terms of $H$. We test for failure across a range of $H_c$ and $T_{ice} = -20$, $-10$ and $-5\,°C$ given normal basal slip conditions ($B_n$). We also test for failure with low ($B_h$; high slip) and high ($B_f$; approaching frozen) basal friction conditions while maintaining $T_{ice} = -20\,°C$. See Methods for further information on the model workflow and parameter assignment.

Viscous deformation consistently leads to ice advance, surface lowering and bulging at the waterline (Fig. 1). This is the physical consequence of longitudinal deviatoric and shear stresses concentrating near the waterline of the calving face, causing focused englacial strain and extrusional flow[5,7,11] (Fig. 1a, b). This viscous deformation imposes tensile stress on the glacier surface that initiates crevassing in HiDEM$_{be}$[19] (Fig. 1c, d). Calving via forward rotation ensues if the surface crevasse penetrates through the glacier's thickness.

Structural ice-cliff failure is observed for $B_n$ conditions when $H_c \geq 136$ m. The retreat of simulated glaciers thinner than this threshold ($H_c \leq 127$ m) is dominated by buoyancy-driven calving, consistent with previous modeling investigations into the spectrum of glacier calving styles[19]. Buoyancy-driven calving, as described in Benn et al.[19], occurs when a glacier terminus is in hydrostatic disequilibrium and buoyancy forces lead to upward rotation, basal fracturing and iceberg calving via outward block rotation. We further detail the differences between buoyancy-driven calving and the modes of structural ice-cliff failure in Supplementary Note 2.

For glacier thicknesses that are susceptible to ice-cliff structural failure, simulations show the 'time to failure' (TtF) decreasing with increasing $H_c$, warming $T_{ice}$, or increasing bed friction (Table 1). This is due to the impact of thickness and temperature on the rate of viscous deformation[23], angle of forward lean (Supplementary Fig. 2) and the stresses that increasingly the tall ice cliffs are subject[4]. The retreat magnitude and TtF are used to calculate time-averaged retreat rate ($\hat{C}$, m d$^{-1}$) in Table 1 and Fig. 2. The influence

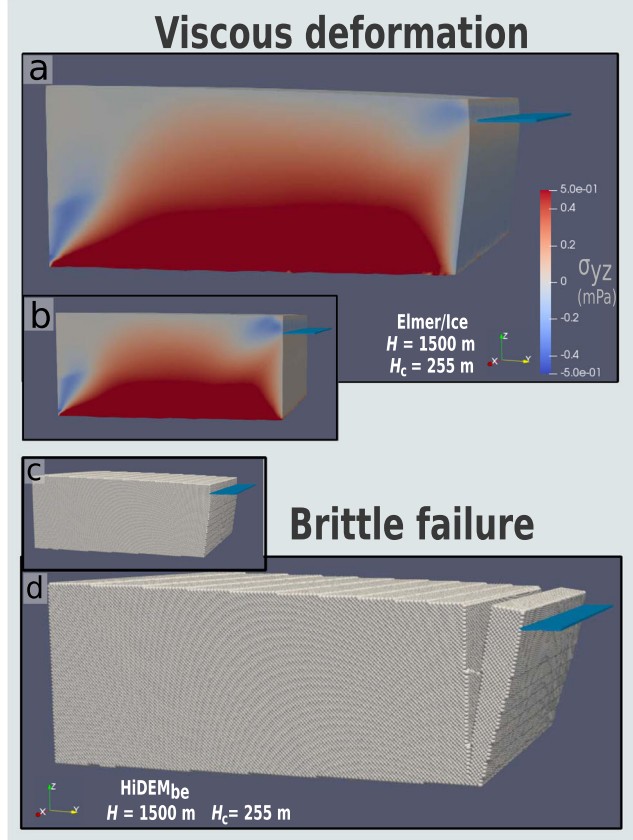

**Fig. 1 Mode of structural failure in which tensile failure follows viscous deformation.** **a** Viscous deformation of a glacier geometry as modeled in Elmer/Ice. The Cauchy shear stress tensor component aligned with flow direction ($\sigma_{yz}$) shows a maximum at the base and waterline of the calving face. **b** The geometry and $\sigma_{yz}$ field shown in (**a**) prior to viscous deformation. **c** The brittle-elastic version of the Helsinki Discrete Element Model (HiDEM$_{be}$) domain initialized with the output geometry of the Elmer/Ice simulation in **a**. **d** An example of brittle failure in HiDEM$_{be}$ caused by tensile stress at the glacier surface leading to surface crevassing and calving via forward block rotation. $H$ = calving face thickness; $H_c$ = ice-cliff height.

of varying $B$, as well as degree-of-buoyancy and surface slope, are discussed further in Supplementary Note 4.

Surface lowering through viscous deformation will alter glacier surface slope and the $H$ and $H_c$ of calving fronts. The magnitude of change will be dictated by a number of ice and environmental conditions (e.g., basal slip, inflow velocity, ice temperature, initial surface slope and calving front thickness). For the standard basal slip conditions used in this study, test simulations are run as examples of how the timing and mechanism of succeeding, secondary calving events will be altered due to viscous deformation. The three considered scenarios are detailed in Supplementary Note 3. In the first scenario, surface lowering results in the newly exposed ice cliffs to no longer be susceptible to structural failure. The newly exposed ice cliffs associated with the other scenarios still structurally fail, though retreat slows or accelerates depending on the evolution of the terminus geometry. Retreat accelerates when progressively taller ice-cliff heights are exposed through each cycle of ice-cliff failure, marking the initiation of MICI.

Proglacial mélange can exert a resistive force on a calving face that, if large enough, may inhibit cliff failure collapse[3,12]. To determine the force required to halt ice-cliff failure, a synthetic

**Table 1 Magnitude and rate of ice-cliff failure resulting from viscous deformation and brittle failure.**

Columns 2–7: $T_{ice} = -20\,°C$; columns 8–13: $T_{ice} = -10\,°C$; columns 14–16: $T_{ice} = -5\,°C$.

| $H/H_c$ (m) | $B_f$ TtF | $B_f$ R | $B_f$ $\hat{C}$ | $B_n$ TtF | $B_n$ R | $B_n$ $\hat{C}$ | $B_h$ TtF | $B_h$ R | $B_h$ $\hat{C}$ | $B_n$ TtF | $B_n$ R | $B_n$ $\hat{C}$ | $B_n$ TtF | $B_n$ R | $B_n$ $\hat{C}$ | Back force (N m$^{-1}$) |
|---|---|---|---|---|---|---|---|---|---|---|---|---|---|---|---|---|
| 800/136 | | | | | | | | | | | | | 33.50 | 45 | 1 | $4.2 \times 10^{6}$ |
| 850/144 | | | | | | | | | | | | | 32.50 | 20 | 1 | # |
| 900/153 | | | | | | | | | | | | | 26.75 | 20 | 1 | # |
| 950/161 | 41.00 | 30 | 1 | 47.25 | 80 | 2 | | | | 40.00 | 40 | 1 | 16.25 | 20 | 1 | |
| 1000/169 | 33.25 | 40 | 1 | 37.25 | 55 | 2 | | | | 36.25 | 60 | 2 | 9.75 | 50 | 5 | $6.6 \times 10^{6}$ |
| 1500/255 | 10.00 | 90 | 9 | 12.00 | 90 | 8 | 29.00 | 190 | 7 | 4.50 | 120 | 27 | 1.75 | 160 | 92 | $1.0 \times 10^{7}$ |
| 2000/339 | 1.50 | 180 | 120 | 1.50 | 240 | 160 | 8.75 | 230 | 26 | 1.00 | 220 | 220 | 0.50 | 320 | 640 | $5.7 \times 10^{7}$ |
| 2500/424 | 0.50 | 350 | 700 | 1.00 | 440 | 440 | 1.75 | 460 | 260 | 0.25 | 360 | 1440 | 0.25 | 370 | 1480 | $2.1 \times 10^{7}$ |

The time to failure (TtF; d), retreat magnitude (R; m) and calving retreat rate ($\hat{C}$; m d$^{-1}$) found with the Elmer/Ice-HiDEM$_{be}$ (the brittle-elastic version of the Helsinki Discrete Element Model) workflow over the range of susceptible ice thicknesses with varying ice temperature ($T_{ice}$) and basal slip conditions. $B_n$, $B_f$, and $B_h$ represent normal, approaching frozen and high basal-slip conditions. The magnitude of mélange back force required to halt failure for the associated $B_n$, $T_{ice} = -5\,°C$ scenario is provided. Blank spaces denote simulations in which failure did not occur within our 50 d viscous deformation window (Supplementary Note 3). Hash marks denote simulations that were not conducted for this study.

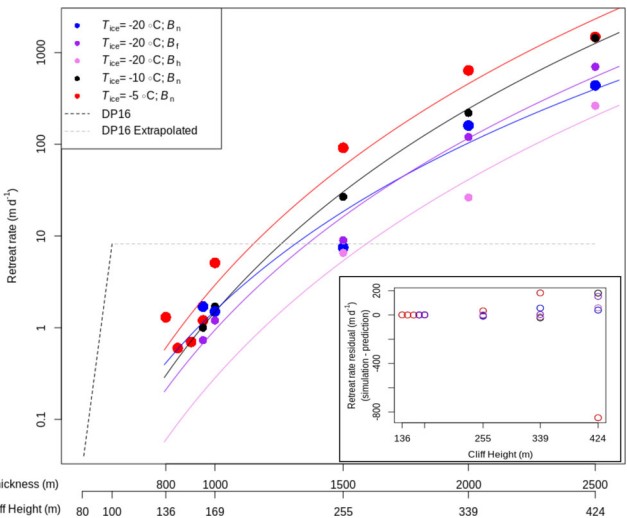

**Fig. 2 Retreat rates resulting from ice-cliff failure as a function of glacier thickness or cliff height.** Points represent results from the Elmer/Ice–HiDEM$_{be}$ (the brittle-elastic version of the Helsinki Discrete Element Model) simulation series, which considers the interaction of viscous deformation and brittle failure. Scenarios with varying ice temperatures ($T_{ice}$) and bed conditions ($B_n$, $B_f$ and $B_h$ representing normal, approaching frozen and high-slip conditions, respectively) are colour coded as follows: $T_{ice} = -20$ °C, $B_n$ (blue); $T_{ice} = -20$ °C, $B_f$ (purple); $T_{ice} = -20$ °C, $B_h$ (pink); $T_{ice} = -10$ °C, $B_n$ (black); $T_{ice} = -5$ °C, $B_n$ (red). Solid lines represent the fitted power-law relationships as per Eq. (1). The dashed line represents retreat rates implemented in DeConto and Pollard[1], with the light grey dashes denoting the maximum retreat rate implemented. Inset: Residuals of power-law fit to retreat rates associated with simulated ice-cliff failure events, with color coding the same as the main plot.

mélange consisting of a plane of unbonded particles that are restricted from moving out of the domain is incorporated in the HiDEM simulations. The resulting depth-integrated force (N m$^{-1}$) along the calving face is calculated from the strain asserted on the exposed particles at the glacier terminus. Incorporating synthetic mélange of varying thicknesses in our idealized simulations shows that the back force required to halt collapse ranges from 4.2 e$^6$ to 5.7 e$^7$ N m$^{-1}$ (Table 1). This value generally increases with $H_c$; a dip in the necessary back force for the greatest tested $H_c$ (424 m), though counterintuitive, is explained by the minimal viscous deformation that is necessary to precondition the towering ice cliff for collapse. This allows the mélange to more efficiently counter the less-concentrated longitudinal stress at the waterline. These calculated mélange back force magnitudes are similar to those known to supress fracturing, calving and iceberg overturn at marine-terminating glaciers[24–28] (Table 1). We expect that greater back force will be necessary to counter increased driving stresses and inhibit structural failure of new calving faces if surface slopes steepen due to viscous deformation leading to the initial calving event.

The simulations described in this section depict one end-member of the spectrum of modes through which structural cliff failure could occur. Shear-localisation dominates at the other end of the spectrum, where either shear strength is reduced or shear stresses increase and the calving face enters a transition region where ductile and brittle deformation simultaneously occur[29].

**Shear localization.** When calving faces are weakened in HiDEM$_{be}$ through decreasing the width of inter-particle beams, decreasing yield strength or increasing initial damage, shear localization leads to shear-band formation, slumping of vertical

calving faces and buoyant uplift (Fig. 3a, b) (see Methods). The narrowing of inter-particle beams ($W$) reduces bending stiffness more quickly ($\sim W^4$) than tensile stiffness ($\sim W^2$), and as beams bend, shear failure is induced in the glacier (Fig. 3a). Increasingly narrow beams are needed to initiate shear in simulations of thinner glaciers. Shear localization is also observed when the initial damage is substantially increased. For example, with the standard yield threshold of 0.2 MPa, shear localization is observed when porosity is increased from 10% (standard) to 60%. When the ice strength is decreased through lowering the yield threshold to 0.1 MPa, shear localization is observed when porosity reaches 50%. Ice-cliff failure for these cases is dominated by a slumping of the surface at the calving face (Fig. 3b).

These simulations show that cliff failure will occur more readily if ice has relatively low shear strength or is damaged and that there will be a transition between tensile and shear failure at a thickness threshold that will be dependent on numerous conditions that influence viscous deformation rates and shear strength. To reach that transition point, cliff failure will need to continually retreat the calving front to locations at which progressively taller ice cliffs are exposed. This will be necessary to counter surface lowering of the terminus due to viscous deformation. It is not possible to validate the parameters that influence shear strength at this point in time due to a lack of relevant observations, and TtF cannot be quantified as a result. Importantly, however, these descriptive simulations highlight the conservative nature of the TtF values reported for the viscous deformation–tensile failure mode above.

**Visco-elastic flow.** Once shear stresses increase so that a calving face enters the transition region between brittle and ductile deformation[29], viscous deformation and fracture can no longer be separated. We simulate such mixed-mode behaviour with HiDEM$_{ve}$, which models both viscosity and elasticity in simplified forms but captures their continuous interaction (i.e., semi-brittle flow). HiDEM$_{ve}$ is used here to assess how viscous deformation and brittle failure interact to form shear bands and influence the evolution of ice-cliff failure and glacier collapse via MICI. Because brittle failure occurs over a much shorter timescale than viscous deformation, the latter process is accelerated when it is incorporated in HiDEM$_{ve}$ simulations. Ice-cliff deformation and failure thus do not interact on realistic timescales in HiDEM$_{ve}$, so it is also not possible to quantify TtF or $\hat{C}$ in these simulations. The HiDEM$_{ve}$ simulations are valuable, however, as they provide insights into how viscous and brittle processes might interact and influence MICI. More information regarding HiDEM$_{ve}$ is found in Methods.

When applied to the previously described geometries with limited basal slip, HiDEM$_{ve}$ simulations show a pattern of waterline bulging and forward lean similar to the terminus evolution seen in Elmer/Ice. However, pronounced shear bands emerge up-glacier of the calving front for the greatest terminus thicknesses, highlighting the importance of the interplay between shear fracture and viscous flow in MICI (Fig. 3c).

A second set of experiments applies HiDEM$_{ve}$ to a domain approximating the grounding line of Thwaites Glacier, West Antarctica, without a fringing ice shelf (see Methods). Dense crevasse formation, calving via block rotation and slumping of the near-terminus ice surface is observed in simulations where basal slip is increased to match the low viscosity of HiDEM$_{ve}$. Failure of thinner domains ($H \sim 880$ m) is again dominated by vertical crevassing and block rotation, the crevassing being induced by tensile stress caused by stretching (Fig. 3d; Supplementary Movie 1). Brittle-compressive failure as described by Schulson[29] causes shear-band formation and slumping to become dominant

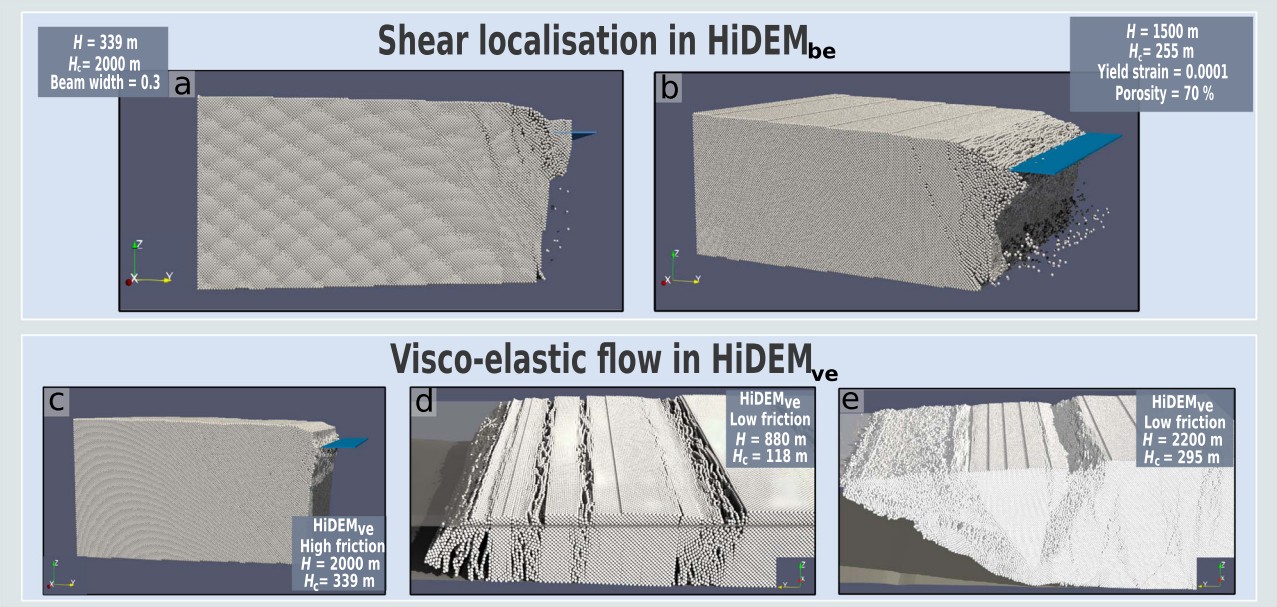

**Fig. 3 Modes of structural failure dominated by shear-band formation.** Shear localization is observed in HiDEM$_{be}$ (the brittle-elastic version of the Helsinki Discrete Element Model) simulations if, given ice-cliff height ($H_c$), inter-particle beam widths are sufficiently narrow (**a**) or the ice is sufficiently weak and damaged (**b**). Damage is represented by porosity (the percentage of pre-broken bonds). **c** Surface slumping and waterline bulging resulting from visco-elastic deformation simulated in the viscous mode of HiDEM$_{ve}$. **d** Vertical crevasses and calving via outward buoyant block rotation at a relatively thin glacier with substantial sliding ($f = 10^{-5}$), as simulated using the intermediate elastic-brittle and viscous mode of HiDEM$_{ve}$. **e** Crevassing, shear bands, slumping and mélange growth were observed as thickness and friction increased ($f = 10^{-4}$) using the same intermediate mode of HiDEM$_{ve}$ as in **d**. Details on f are included in Supplementary Note 7. H = calving face thickness. Blue and white transparent planes represent the waterline.

characteristics of the terminus collapse as the simulated glacier domain increases in thicknesses ($H \sim 2200$ m), which results in the rapid formation of a very dense mélange (Fig. 3e).

Through an assessment of kinetic energy, we find a slight acceleration of ice-cliff failure via shear-band formation with increasing ice thickness in the HiDEM$_{ve}$ simulations (Supplementary Fig. 4). However, the relationship is relatively weak since, in this case, glacier deformation is dominated by surface slumping and the formation of crevasses and shear bands. This contrasts with the strong dependence of TtF on $H_c$, as seen in the results of the viscous deformation–brittle failure simulation series. In that case, deformation is dominated by viscous flow as described by Glen's flow law, with the deformation rate increasing by $\sim H^3$. The relationship between the timing of cliff failure and ice thickness for that mode of failure in our simulation series (TtF $\sim 1/H^4$) is considered reasonable as the weight of the developed overhang also grows in proportion to $H$ and $H_c$, further hastening failure. These differences indicate a qualitative transition in the dominant failure mode at a cliff height above which shear bands begin to form. A sharp decline in TtF is expected when this transition ($H_c \sim 200$ m) is reached. At the same time, the process generates a thick and dense mélange in front of the calving face, the height of which has been lowered by slumping action. This has the potential to stabilise the glacier terminus (Fig. 3e, d).

**Calving rate parameterization.** In Eq. (1) we propose a simple and conservative ice-cliff failure rate parameterization in which $\hat{C}$ (m d$^{-1}$) is represented as a fitted power-law function that is based on $H_c$ and dependent on $T_{ice}$ and $B$. Equation (1) is fitted to the values of $\hat{C}$ that were derived from the simulation results of the viscous deformation–brittle failure ice-cliff failure mode simulated by the Elmer/Ice–HiDEM$_{be}$ workflow (Fig. 2). We set Eq. (1) for $H_c > 135$ m based on the minimum ice-cliff height at which failure occurs, which is found through the Elmer/

Ice–HiDEM$_{be}$ simulation series.

$$\text{For } H_c > 135 \text{ m} \quad \hat{C} = I(H_c)^\alpha \quad \begin{aligned} T_{ice} &= -20\,°C; \; B_f|I = 3.7e^{-16}; \; \alpha = 6.9 \\ T_{ice} &= -20\,°C; \; B_n|I = 5.1e^{-14}; \; \alpha = 6.0 \\ T_{ice} &= -20\,°C; \; B_h|I = 3.2e^{-17}; \; \alpha = 7.2 \\ T_{ice} &= -10\,°C; \; B_n|I = 6.9e^{-17}; \; \alpha = 7.3 \\ T_{ice} &= -5\,°C; \; B_n|I = 1.9e^{-16}; \; \alpha = 7.3 \end{aligned}$$

$$(1)$$

Due to the incongruent timescales of viscous and elastic processes in HiDEM$_{ve}$ and unvalidated settings for the parameters controlling shear strength in HiDEM$_{be}$, the impact of shear-band formation is not included in the retreat rate parameterization at this time. Therefore, values of $\hat{C}$ derived from Eq. (1) are considered conservative, since shear-band formation and damaged ice with lower effective viscosities will promote deformation and calving, ultimately quickening failure and enhancing retreat rates[7,30,31].

The residuals of the fitted ice-cliff failure retreat rate parameterization are small for the lower-range of cliff heights that failed in our simulations (Fig. 2, inset). Fit decreases as cliff height increases, with large absolute errors seen for individual simulations of tall cliffs. Noteworthy situations include: (1) the outlying, 850 m d$^{-1}$ residual in Fig. 2 (inset) for the scenario resulting in the quickest retreat ($H_c = 424$ m, $T_{ice} = -5$C, $B_n$) and (2) the 660 m d$^{-1}$ error between the retreat rate predicted by our parameterization in comparison to that associated with the run-away retreat, secondary calving event scenario ($H_c = 347$ m, $T_{ice} = -5$C, $B_n$) described in Supplementary Note 3. Both of these situations are associated with very quick failure (TtF < 24 h) and the corresponding errors are largely influenced by the 6-h temporal resolution of our Elmer/Ice simulation. The errors in predicted retreat rates associated with the two secondary cliff failure simulations are of the same magnitude as the residuals of the retreat rate parameterization, which was fit to the initial

ice-cliff failure simulation results. In addition, the derived values of $\hat{C}$ for these simulations correctly fall along the trendline of their respective temperature and basal slip scenario.

Although large model variance is apparent for some simulations of individual structural ice-cliff failure events, the presented retreat rate parameterization is a contribution to the ice-sheet modeling community that is concerned with adequately representing trends of retreat[4]. Our fitted retreat rate parameterization captures trends in retreat rates with increasing thickness, warming ice temperature and altered basal slip and the non-linear increase in $\hat{C}$ with increasing $H_c$. The latter relationship points to the potential for alarmingly rapid retreat and the initiation of run-away MICI retreat if viscous deformation and/or mélange buttressing do not counteract ice-cliff instability. It is imperative that the retreat rate parameterization proposed here be used in conjunction with model representation of ice surface lowering via dynamic thinning at a sufficiently high temporal resolution, which will determine whether the competing processes of ice flow and structural ice-cliff failure lead to either glacier stability or runaway MICI retreat.

Uncertainty in Eq. (1) arises from parameter assignment, model set-up and the resolution of our idealized simulations. These factors, which can be considered if the retreat rate parameterization is refined in the future, will influence the quantitative model results but will have a limited effect on the qualitative behaviour of our derived relationships. The conservative property of Eq. (1) is further supported by the accelerating effect of melt undercutting and shear-band formation. Test simulations also show that adjusting $H$ to floatation for a given $H_c$ will accelerate failure due to the increasing longitudinal stresses associated with the thicker glacier (Supplementary Note 5). Values of $\hat{C}$ are also conservative in terms of HiDEM$_{be}$ model structure and resolution (Supplementary Note 5). Furthermore, this retreat rate parameterization was derived for simulations of largely undamaged ice with high fracture toughness. Weaker ice would cause failure to initiate at lower cliff heights and alter the curve placement in Fig. (2) (Supplementary notes 1 and 5).

The calving retreat rate parameterization presented in Eq. (1) represents a specific mode of ice-cliff failure. Distinct calving parameterizations should be implemented to represent calving processes for ice thicknesses below the structural instability threshold, as previously proposed by Schlemm and Levermann[12]. A final challenge for the ice-sheet modeling community will be to determine how to incorporate mélange back force, which could potentially stabilise an ice cliff, in continental-scale models[32,33], this being an aspect in which the calving retreat rate parameterization is not conservative. We provide the values of back force required to inhibit ice-cliff failure for idealized scenarios, and note that these values can be introduced through different combinations of mélange thickness and rigidity, jamming against bathymetric highs, and lateral resistance through contact with fjord sidewalls.

**Implications for glacier and ice-sheet retreat.** Climate change is transforming the Antarctic continent. Expected ice-shelf loss and glacier retreat could expose tall and unstable ice cliffs, putting parts of the region at risk of ice-cliff failure, MICI initiation and accelerated ice loss[9,34]. Our idealized simulations illustrate modes of structural cliff failure that emerge from combinations of ice thickness/cliff height, basal slip, strength and pre-existing damage. Glaciers with undamaged ice that is strong in shear and exposed to limited basal slip are shown to characteristically deform through surface lowering, waterline bulging and cliff advance. The resulting overhang imposes tensile stresses that can cause collapse via brittle failure, leading to iceberg calving that is

distinct from buoyant calving[19,35]. Glaciers that are subject to greater basal slip and enhanced stretching develop tensile crevasses that lead to calving events. Shear stress becomes the dominant contributor to the failure of thicker glaciers, causing failure via shear-band formation and slumping. This supports explanations that vertical compression and slumping of the upper surface could contribute to ice-cliff failure[5,7].

Our results from the visco-elastic model show divergent behaviour of kinetic energy release and terminus retreat as ice thickness increases. Thus, for all modes of failure (viscous deformation followed by brittle failure, shear localization and visco-elastic deformation), ice fronts quickly reach states of rapid failure with increasing cliff heights. In all cases, ice-cliff failure produces copious mélange, either in the form of individual icebergs or, in the case of the visco-elastic model, a highly fractured mélange shelf. This mélange could provide sufficient resistance to suppress further calving, unless it can be evacuated away from the ice front. Therefore, the rate of mélange evacuation is a rate-limiting process for MICI, and will be influenced by the particular embayment or fjord geometry[12]. In wide embayments, the presence of bathymetric highs will be crucial for determining whether mélange can be evacuated. This is the case for Thwaites Glacier, which holds ~0.6 m of global potential SLR[36], is vulnerably located above an over-deepening basin[8] and has thicknesses at which structural failure could occur within a few kilometres of its current grounding line. Further work is now needed to understand how MICI could unfold at Thwaites Glacier and other locations, with consideration given to the potential for mélange to build-up and provide sufficient back force to inhibit calving. It is also important to test how different sliding laws will impact model results.

Structural ice-cliff failure was represented as a simple, saturating horizontal wastage rate (0–3 km year$^{-1}$) in simulations of future Antarctic Ice Sheet retreat by DeConto and Pollard[1]. While we report lower values of $\hat{C}$ than DeConto and Pollard[1] for 135 m < $H_c$ < ~ 200 m, our reported values of $\hat{C}$ for greater cliff heights agree with previous assessments regarding the potential for more rapid ice-cliff failure retreat rates to be realised[1,7,31]. This is reasonable, as our retreat rates are relatively similar for the previously considered values of $H_c$ (80–100 m) (Fig. 2), but our simulation series then extends far past these heights. Uncertainty exists in how fast ice sheet retreat will ultimately occur, as numerous other processes, including marine ice-sheet instability (MISI) and mélange buttressing, will influence rates of ice-cliff failure and if or when run-away MICI retreat will initiate. Ice sheet modeling studies are the next step in determining this. Crucially, these studies must assess how MICI and MISI will interact, as these processes could vastly reconfigure the Antarctic coastline.

## Methods
We utilise a suite of high-fidelity glacier models to (a) investigate the influence of viscous deformation, brittle failure and shear-band formation on ice-cliff failure and MICI retreat, (b) derive an ice-cliff failure retreat rate parameterization and (c) quantify the magnitude of back force that is needed from mélange to suppress ice-cliff failure.

**Elmer/ice.** The viscous deformation and advance of simulated glaciers are conducted with the open-source, full Stokes continuum glacier model, Elmer/Ice[15]. We use a 4 km (long) × 3 km (wide) model domain with a surface slope of 0.021 and a retrograde bed slope of −0.018. These slopes are based on the setting of Thwaites Glacier, West Antarctica. Because it is impractical to model the entirely of the vast Thwaites Glacier calving front in HiDEM, we use passive sidewalls to represent conditions along the calving front that are not greatly influenced by lateral resistance from sidewalls. Noise is introduced to the basal surface to remove unphysical stress concentrations when Elmer/Ice output was transferred to HiDEM$_{be}$. All simulations are initialized with a linear bed friction law[37]:

$$\tau_b = Cu_b \qquad (2)$$

where $\tau_b$ is basal shear stress, $u_b$ is the glacier sliding velocity and $C$ is the friction coefficient[20]. As we are interested in exploring the parameter space in which structural failure of ice cliffs occurs, we do not allow the geometry to relax in our simulations; instead, we find the result of varying a number of input factors on the timing and magnitude of retreat associated with ice-cliff failure.

Elmer/Ice runs are conducted with a 6 h time step. We test the sensitivity of our results to the Elmer/Ice time step by comparing the time to failure (TtF) for simulations of a 2500 m thick, $-10\,°C$ glacier run at 1 and 6 h time steps, TtF being the duration of time needed for viscous deformation to precondition the glacier to collapse via brittle failure in HiDEM$_{be}$. For this fast-deforming scenario, the difference in TtF was 1 h, which is within the temporal uncertainty associated with the majority of the model simulations run at a 6 h time step.

**HiDEM$_{be}$.** Brittle failure, after viscous deformation in Elmer/Ice, is simulated with the standard brittle-elastic implementation of the 3D Helsinki Discrete Element Model (HiDEM$_{be}$)[17,18,38]. HiDEM$_{be}$ replicates a wide range of calving styles[19] and successfully reproduced observed calving events at Kronebreen, Svalbard[22], Totten Ice Shelf, Antarctica[21] and Bowdoin Glacier, Greenland[20]. The fracture properties used in this study are informed by those used to model Bowdoin Glacier[20] and Store Glacier (Joe Todd, pers. comm.). Utilising HiDEM$_{be}$ allows us to explicitly simulate fracture, avoiding assumptions of when and where calving occurs based on simulated stress fields in continuum ice flow models[21].

This numerical calving model represents glaciers as assemblages of densely-packed, discrete particles bonded by elastic, breakable beams[22]. More information on the structure of HiDEM is provided in Supplementary Note 1 and van Dongen et al.[20] Unless otherwise noted, our simulations were built with 20 m particles, a critical strain threshold of 0.0002 and seeded with 10% broken bonds (the 'porosity' parameter) to represent pre-existing micro-fractures within the ice[18,21] (Table 2). Bonds fracture when the critical strain threshold is exceeded, with the 0.0002 value falling within the range of strain at which brittle ice fractures under tension[17–19,21,29].

The lattice structure and standard parameter settings (e.g. inter-particle beam width, strain threshold and number of seeded broken bonds) effectively enhances fracture toughness and discourages shear localisation and damage at the stresses induced in our HiDEM$_{be}$ simulations. We therefore use HiDEM$_{be}$ to simulate tensile failure after viscous deformation. Test simulations of shear failure in HiDEM$_{be}$ after parameter adjustment are conducted to investigate the role of shear-band formation in cliff failure and are described below.

**Elmer/Ice–HiDEM$_{be}$ workflow.** Evolved Elmer/Ice glacier geometries are exported to HiDEM$_{be}$ to test for cliff failure. This workflow allows us to consider the contrasting fluid and brittle-solid characteristics of ice[19,22]. Key model parameters are included in Table 2. Using a Newtonian algorithm we test Elmer/Ice output for collapse in HiDEM$_{be}$ at varying time-steps of a given simulation to determine the TtF. The associated retreat magnitude is measured in Paraview (v. 5.5.0).

A series of simulations are run to test how TtF and retreat magnitude are influenced by glacier thickness ($H$) (with a corresponding cliff height ($H_c$)), ice temperature ($T_{ice}$) and basal friction ($B$). Grounded glaciers with $H = 1000$, 1500, 2000 and 2500 m are tested for cliff failure over 50 d of viscous deformation occurring at $T_{ice} = -20$, $-10$ and $-5\,°C$. Extra simulations are conducted between $H = 650$ and 950 m to determine the thickness, within 50 m, at which cliff failure initiates for a grounded glacier. These simulations are considered as normal ($B_n$) with a slip coefficient prescribed in Elmer/Ice that decreases as a linear function from 1.0e2 to 1.0e4 m a$^{-1}$ MPa$^{-1}$ along the 4 km model domain and a surface-to-base velocity ratio of 2.6 for $H = 1500$ m. Inflow velocity was set at 1500 m year$^{-1}$,

so basal slip and inflow are broadly similar. This initialization results in velocity values of ~1.5–2 km year$^{-1}$ up-glacier of the calving front. To assess how surface lowering through viscous deformation will impact subsequent calving events, test simulations of the succeeding calving event after the initial cliff failure are run for $H = 1000$ ($T_{ice} = -5\,°C$) and 2000 m ($-5$ and $-20\,°C$). For these simulations, Elmer/Ice is initialized with the surface slope and $H_c$ of the new calving face after the first calving event simulated in HiDEM$_{be}$. Evolved geometries are again transferred from Elmer/Ice to HiDEM$_{be}$ to test for failure.

To test the impact of varying basal slip on the timing and magnitude of cliff failure, simulations were conducted over the above range of thicknesses with the basal slip coefficient decreased and increased by an order of magnitude. Increasing basal slip conditions ($B_h$) results in a low surface-to-base velocity ratio (equal to 1.5 for $H = 1500$ m) and accelerated overhang development, as basal advance is more similar to advance of the glacier surface. Decreasing slip so that the bed approaches a frozen condition ($B_f$) results in simulation conditions where the surface-to-base velocity ratio is high (5.7 for $H = 1500$ m) and the overhang development is slowed, as surface advance outpaces that of the glacier base. The inflow velocity was adjusted to 1250 and 1750 m year$^{-1}$, for simulations with $B_h$ and $B_f$ conditions. $T_{ice}$ remained static at $-20\,°C$ for these simulations, this temperature representing $T_{ice}$ at Thwaites Glacier in the vicinity of the waterline where longitudinal deviatoric and shear stresses concentrate[39].

To test changes to a glacier's degree of buoyancy, we increase the waterline to the flotation point for $H = 1500$ m glacier, effectively decreasing the cliff height by 26%. We also test the TtF and failure mode for a glacier at buoyancy with a 255 m cliff, which is the original cliff height associated with the grounded, $H = 1500$ m case. This effectively increases the total glacier thickness by 34%. $T_{ice}$ is assigned $-20\,°C$ for these sensitivity tests.

Mélange can inhibit calving by asserting back force on a glacier terminus[12,24,40]. We determine the back force required from mélange to impede cliff failure by varying the thickness of a synthetic mélange plane in simulations of well-grounded glaciers with $H = 800$, 1000, 1500, 2000 and 2500 m that are viscously preconditioned for collapse. Inter-particle bonds and 10% of the mélange particles are removed to allow vertical and horizontal movement of the mélange. The mélange abutted a bedrock 'wall' at the end of the fjord so mélange particles cannot be evacuated from the fjord by the failing calving face. The depth-integrated mélange force, or back force per horizontal unit, is calculated from the strain asserted on particles associated with the glacier calving face. The depth-integrated mélange force is reported in N m$^{-1}$. We report the average, depth-integrated mélange back force over the entire calving face in Table 1 and in the text.

The standard HiDEM$_{be}$ model set-up does not exhibit shear-band formation under the compressive stresses produced in our idealized simulations. We vary the yield strength, porosity and beam width parameters to assess what conditions of ice strength and internal damage conditions, respectively, lead to shear-band formation in HiDEM$_{be}$. Yield strength and porosity test simulations were conducted for the $H = 1500$ m, $T_{ice} = -20\,°C$ scenario. The porosity parameter represents damage and is the percentage of initially broken bonds in a HiDEM simulation. For yield strain of 0.0002 and 0.0001, the latter representing the minimum tensile strain threshold of ice[29], porosity is increased by 10% until complete failure via shear-band formation is observed.

The width of inter-particle cylindrical beams ($W$) impacts HiDEM$_{be}$ tensile and shear strength. Test cases were run for $T_{ice} = -5\,°C$ and $H = 1000$, 1500 and 2000 m to demonstrate how weakening shear strength, relative to tensile strength, influences failure. Tests are run with $W = 0.55$, 0.4 and 0.3. The default setting for $W$ is 0.7, this being a non-dimensional relation between particle diameter and beam width. This parameter assignment has been validated in simulations of calving at marine-terminating glaciers in and Svalbard[25] and Greenland[20] (Todd et al. (in prep)).

**HiDEM$_{ve}$.** In HiDEM$_{ve}$, solid blocks are connected by breakable beams in the same way as in HiDEM$_{be}$ to capture the brittle-elastic component of ice deformation. To adapt the model to viscoelasticity, adjacent particles in HiDEM$_{ve}$ also interact through an added viscous potential. This is a short-range cohesive force between adjacent particles within a specified range. The viscous force only has a radial component that mimics the typical behaviour of fluids by having a strong resistance to volume change but a weak resistance to irreversible shear deformations. The deformation rate is determined by the shear stress. These adaptations allow the model to be tuned to simulate pure brittle-elastic and pure viscous deformation.

Here we use HiDEM$_{ve}$ to explore the role of brittle viscoelastic shear-band formation in cliff failure and MICI retreat. Simulations of low slip glacier conditions that use HiDEM$_{ve}$ in its most viscous form use the same $4 \times 3$ km model domain as those used in the Elmer/Ice–HiDEM$_{be}$ workflow. An intermediate brittle visco-elastic version of HiDEM$_{ve}$ is used to investigate shear-band formation for scenarios with basal slip scaled down to match the low viscosity of HiDEM$_{ve}$. This version is applied to a 3D-extruded flow-line profile of Thwaites Glacier that is cut slightly upstream of the grounding line, where $H$ is approximately 900 m, to create a hypothetical MICI-initialization scenario. The observed level of the sea surface is used in these simulations. The thickness of the glacier is changed by applying a scaling parameter, SCL, to the vertical component of the ice surface and sea level. We compute the total kinetic energy of the glacier as a function of time for SCL = 0.75, 1.0, 1.25, 1.5, 1.75, 2.0 and 2.5. We estimate the viscosity of HiDEM$_{ve}$ via the

---

**Table 2 Key parameters in Elmer/Ice–HiDEM$_{be}$ offline model coupling workflow used to derive the ice-cliff retreat rate parameterization.**

| Model | Parameter | Value |
|---|---|---|
| Elmer/Ice | Glen exponent | 3 |
| | Prefactor (s$^{-1}$ Pa$^{-3}$) | 3.985e13 ($T_{ice} \leq -10\,°C$) |
| | | 1.916e3 ($T_{ice} > -10\,°C$) |
| | Activation energy (KJ mol$^{-1}$) | 60 ($T_{ice} \leq -10\,°C$) |
| | | 1139 ($T_{ice} > -10\,°C$) |
| | Gas constant (J mol$^{-1}$ K$^{-1}$) | 8.314 |
| | Slip coefficient | 1.0e2 to 1.0e4 m a$^{-1}$ MPa$^{-1}$ |
| | Ice temperatures (°C) | $-5$, $-10$, $-20$ |
| | Timestep (hr) | 6 |
| HidEM$_{be}$ | Yield strength (MPa) | 0.2 |
| | Porosity (%) | 10 |
| | Young's modulus (Pa) | 1.0e9 |
| | Particle size (m) | 20 |
| | Timestep (s) | 1.0e-4 |

strain-rate of viscous deformation to be of the order $10^{-4}$ to $10^{-5}$ times lower than the real glacier and bed friction was scaled-down accordingly.

**Energy scaling**. When basal friction is scaled down in the viscoelastic simulations to match the low viscosity of HiDEM$_{ve}$, the combination of sliding and flow induces crevassing and shear-band formation substantially up-glacier from the calving face. It is not possible to clearly differentiate between locations that have undergone brittle failure or calving versus viscous deformation, as the two become heavily intertwined when modeling viscoelastic glacier behaviour in HiDEM$_{ve}$. As a result, we cannot properly define the retreat rate ($\hat{C}$) and TtF for the viscoelastic cases. We therefore use kinetic energy as a measure for ice-cliff failure activity. We hypothesise that, as calving and fracture begin, the acceleration of kinetic energy can be described by a single scaling function of time, $g(t)$. The dependence of the kinetic energy on glacier thickness (SCL) and basal friction ($f$) can then be described by rescaling time and energy: $t \Rightarrow t/(\text{SCL}_t f^{S_t})$, and $E_{kin} \Rightarrow E_{kin}/(\text{SCL}_e f^{S_e})$, where SCL$_e$, and SCL$_t$ are functions of SCL.

**Derivation of calving rate parameterization**. Log-log plots are used to determine if a power-law relationship existed between $\hat{C}$ and $H_c$. A power-law equation ($y = Ix^\alpha$) using the common logarithm (i.e., base 10) is fit for this relationship for the results of the $T_{ice} = -5, -10$ and $-20$ °C series of simulations for $B_n$. Fits are also derived for $B_f$ and $B_h$ for $T_{ice} = 20$ °C. We report the fitted values of $I$ and $\alpha$ in Eq. (1).

## Data availability
All information necessary to initialize the idealized model simulations in the utilised open-source models, HiDEM and Elmer/Ice, is found in the Methods. The data generated from the described idealized simulations used to fit Eq. (1) is presented in Table 1. Model initialization scripts and other derived data reported in this article are available from A. Crawford upon request.

## Code availability
Elmer/Ice and HiDEM are both open-source models that are available through their respective online repositories: github.com/ElmerCSC/elmerfem and github.com/joetodd/HiDEM.

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

## Acknowledgements
This work is from the DOMINOS project, a component of the International Thwaites Glacier Collaboration (ITGC). DOMINOS is supported by the Natural Environment Research Council (NERC: Grant NE/S006605/1). This article represents ITGC Contribution No. ITGC-020. The model simulations were conducted with computational resources provided by NERC and PRACE.

## Author contributions

D.B., A.J.C. and J.A. conceived and designed the study. A.J.C. and J.A. conducted the model simulations. J.T. provided guidance regarding Elmer/Ice and HiDEM behaviour and code modification as well as analysis scripts. All co-authors (A.J.C., D.B., J.T., J.A., J.B. and T.Z.) were involved in the assessment of model output. A.J.C. and J.A. derived calving rate equations and energy-dissipation relationships. A.J.C. led the writing and production of figures for the paper. J.A. contributed text and figures, and all co-authors (A.J.C., D.B., J.T., J.A., J.B. and T.Z.) assisted with the paper's development and editing.

## Competing interests

The authors declare no competing interests.
