## [Peer Review File · Nature Communications]

REVIEWER COMMENTS

Reviewer #1 (Remarks to the Author):

This manuscript documents an important modeling investigation into the physical processes driving iceberg calving from tall ice cliffs (commonly referred to as the Marine Ice Cliff Instability or MICI). Using three state-of-the-art models (separately and coupled with each other), the authors identify three different modes of calving from tall marine ice cliffs: viscous deformation leading to brittle fracture, shear band formation, and brittle tensile failure. They construct a parameterization based on the rate of brittle fracture driven by viscous deformation and separately a scaling law for the kinetic energy of calving events. Overall, this study explores an important topic that has been the focus of considerable debate within the glaciology and ice sheet modeling community over the last several years and sheds light on the processes that might be driving the hypothesized ice cliff failure. A completed manuscript would likely be of interest to a broad community of ice sheet modelers and glaciologists.

My major concern regards the validity of the parameterization determined at the end of the manuscript. Clearly a goal of this project is to derive a physically-based MICI parameterization (which has not been available previously, since the existing parameterization is based on few observations). However, to go from the simulations described in this study to a parameterization is a large leap that requires making a number of assumptions about how the simulations described here translate to large-scale viscous ice sheet models. In particular, the coupled ELMER-HiDEM simulations begin with an idealized square terminus geometry, which is then allowed to viscously deform, with periodic "checks" for fracture through simulations with the brittle version of HiDEM. Once the first full-thickness calving event happens, the simulation is stopped and the timing and size and the calving are converted into an "effective" calving rate (C_{hat} in table 1) which is fitted to determine the MICI parameterization. The critical assumption here is that the next calving event that occurs after this first simulated event will continue to follow this effective calving rate, even though it is starting from a very different terminus geometry and stress state (with the separate issue of the potential influence of the calved iceberg on subsequent stress state through a melange-like buttressing effect). The authors have extrapolated from a single simulated calving event on time scales of hours to days to an effective calving rate that would be implemented in ice sheet models through the implicit assumption that many, many calving events (occurring over the time scale of a single viscous model time step of many weeks to months) would continue to occur with the frequency dictated by the initial idealized geometry. I just don't see how this is a physically meaningful or well-supported assumption. Perhaps the authors have a good explanation for why this is justified, but I don't find it in the current version of the manuscript. One possible way to justify this would be to do a long run of the coupled ELMER-HiDEM simulation, with many calving events and a continuously viscously deforming glacier geometry and show that all the calving events follow a calving rate that falls on the parameterized relationship defined in this study. This would certainly be very convincing to showing that this is a robust parameterization that is not too dependent on the pristine initial conditions.

Another issue I have arises from the "threshold" at which MICI starts (Line 90-92). How can we qualitatively distinguish between buoyancy-driven calving and the supposedly MICI-driven calving seen at $H_c > 136$ meters? Both involve full-thickness calving events. Why should one be included in the MICI parameterization and the other not? This makes the threshold of 136 meters seem somewhat arbitrary. Presumably this version of the parameterization would only make sense to implement in an ice sheet model that has a different parameterization for the calving that occurs below this threshold. How does that parameterization know when to stop calving?

A lot of science is being compressed into a short manuscript, so I think the authors can work on the clarity of the manuscript and figures - more suggested edits in this direction can be found below.

Line 23: basins more than 1 km deep [since MICI wouldn't necessarily occur just in any overdeepened

basin]

Line 34: not sure what mounting means here

Line 35: threshold being exceeded at a cliff height of...

Line 38: impossible is a strong word here, especially since you couple ELMER and HiDEM in a way that has the potential to be used for continental-scale simulations, as shown in Memon et al. 2019 including several of the co-authors on this study.

Line 41-42: Not sure I follow the logic of this sentence

Line 63: delete comma

Line 100-101: Could you discuss more about how does this melange result relate to your other results? You've shown here that the likelihood of MICI-like calving is sensitively dependent on the stresses at the terminus. So, how will the glacier geometry and resulting effect on stress state (particularly when the terminus is not a perfect rectangle) change after the first calving event in your simulations? (This is related to my main point above)

Line 145: Im a bit confused about why you use kinetic energy here, when you look at the actual magnitude of calving events in the previous simulations. I agree the kinetic energy is a useful metric, but why change between metrics halfway through the manuscript?

Line 167-174: this discussion of the viscous controls on Ttf would make a lot more sense in the previous section when you are actually talking about these simulations.

Line 174: Something else that is unclear to me here is that if the viscous mechanism dominates (but is fast) at cliff heights of 135-200 m, and then shear bands begin to dominate for $H_c > 200$ m, would we expect that the shear bands would never occur in practice since the viscous mechanism should always occur first? I think I'm confusing myself about this, but this bears discussing I think.

Equation 2: I'm a bit confused about how you got this from table 1 (particularly the gray shaded cells), which shows that no MICI-like collapse occurs for $H_c < 153$ meters and $T_{ice} = -20$ or -10 C. Shouldn't I be set to zero at these heights?

Line 235: Yes, you get higher wastage rates than DP16, but you also have a higher threshold for the onset of MICI calving and lower rates for cliffs below 200 meters (in Figure 2). Shouldn't this be pointed out?

Figure 1: There is a lot happening in this figure (could be split up), and the formatting is inconsistent between panels and backgrounds.

Figure 3c: Not sure what this panel is showing.

Figure 4: Not sure what the y axes are showing in this figure. Also the y-axis of panel b overlaps with panel a.

Table 1: Why not also report the result of simulations for $H_c < 136$ meters? Does calving occur in any of these (even if it isn't MICI calving, see main comment above).

Reviewer #2 (Remarks to the Author):

Marine Ice Cliff Instability (MICI) is currently a hot topic in Antarctic glaciology. In two papers (Pollard et al, 2015 and DeConto and Pollard, 2016) a parameterisation of this mechanism was introduced, based on earlier work of one of the co-authors of this manuscript (Bassis and Walker, 2011). MICI was introduced to enable to simulate Pliocene high sea-level stands that can only be explained by large mass losses from the Antarctic ice sheet. The introduction of MICI in DP16 (DeConto and Pollard, 2016) has profoundly shaken the modelling community, as the mechanism potentially results in future sea-level rise estimates of almost an order of magnitude larger compared with other studies. Therefore, a better comprehension of this mechanism is urgently needed, which this manuscript by Crawford et al. tries to do.

This novel study investigates MICI (Marine Ice Cliff Instability) collapse for brittle and viscous processes (both separated and mixed) using two different ice flow models, i.e. Elmer/Ice for viscous

deformation, and HIDEM for brittle deformation. For undamaged ice, viscous deformation is needed to initiate cliff collapse (by forming a fore bulge), but viscous deformation also imposes tensile stress on the glacier surface, which initiates crevassing (and damages the ice). The HIDEM model is used for to purposes: (i) to investigate cliff collapse for undamaged ice (HIDEM_be) and for damaged ice through shear-band deformation (HIDEM_ve). Based on the result from the undamaged ice experiments, the authors manage to establish a new calving law based on brittle and viscous deformation , which is compared to calving laws for MICI as parameterised by DP16. They furthermore look at the back force of ice mélange that may lead to calving face stability.

The paper is important for providing the reader with new derived calving laws based on MICI, but their use are limited to the undamaged case. These calving rates are also less important than those used in DP16 and cliff failure generally occurs for higher cliffs than the range in DP16. It is a pity that effects of shear failure cannot be accounted for in these laws, as for damaged ice the authors limit this effect by representing kinetic energy as a function of cliff height. In the discussion, the authors make the point that shear band formation may well increase the calving rate (comparable to DP16, but not quantified other than through kinetic energy) and that on the other hand ice mélange formed by the calved ice may exert sufficient large stresses to stabilise the cliffs again.

Major comments

Based on the HIDEM_ve experiments, a further quantification beyond the energy scaling, i.e., providing cliff retreat rates as a function of cliff height, cannot be 'properly' done. If it cannot be done properly, could an estimate or range be given? Because this would underpin further statements on the comparison with parameterisations of DP16.

The same accounts for the effect of ice mélange. In Fig. 3 an example of one of the experiments is shown, but how does this summarise for all experiments?

The paper is quite technical in nature and could benefit from more context. For instance, why is MICI important and why it is controversial; reference to Edwards et al (2019) re: difficulties in reconciling with other SLR estimates; why MICI has been introduced for explaining Pliocene SLR, ... That context is necessary for a publication in a journal targeting a wider public.

The paper also mixes up tenses. Most of it is written in the past tense, but the discussion is in the present tense. For clarity and ease of flow it would be preferential to write everything in present tense.

Detailed remarks

Line 24: I don't know any other ice sheet model than Pollard et al. (2015) and DP16 who have included MICI in simulations. In my mind, only one model uses simple parameterisations, and the other models don't include MICI.

Line 34: explain 'mounting glaciostatic stress'

Line 64 (and others): surface-to-base velocity ratios instead of surface : base

Line 174: reached.

Line 178: remove period after 'equation'.

Lines 191-193: It should be added that the ice mélange could have the opposite effect to put the

'conservative' measure in perspective.

Line 213: the opening sentence is weird. I wouldn't even call it 'new', as it may well have occurred in the past (e.g., Pliocene).

Line 214: 'precarious situation' is not a scientific term to describe what the effect of ice shelves can be. To give a bit more context here, a recent paper by Sun et al. (2020; Journal of Glaciology) has demonstrated in a multi-model context that loss of ice shelves due to MISI (and without MICI) leads to multi-meter SLR on short time scales (hundreds of years). One could therefore argue that key in the stability of WAIS is the point at which ice shelves collapse due to atmospheric/oceanic forcing, more so than the fact whether MICI kicks in or not.

Line 365: It should be stated why a 4 by 3 km domain with passive sidewalls is representative for Thwaites glacier that is much wider than the domain used.

Line 371: C is a friction coefficient. Furthermore, a linear bed friction law is used, while several studies show that basal friction is probably more plastic. What would be the effect of nonlinear sliding on calving rates or do they not have an effect?

Line 481: cases.

Reply to Reviewers: Manuscript #2019-125 by Crawford, Benn, Todd, Åström, Bassis and Zwinger:
“Marine ice-cliff instability modeling shows mixed-mode failure and yields calving rate law”

Original reviewer comments are in black.

Author replies are in blue.

Reviewer 1:

General comments:

This manuscript documents an important modeling investigation into the physical processes driving iceberg calving from tall ice cliffs (commonly referred to as the Marine Ice Cliff Instability or MICI). Using three state-of-the-art models (separately and coupled with each other), the authors identify three different modes of calving from tall marine ice cliffs: viscous deformation leading to brittle fracture, shear band formation, and brittle tensile failure. They construct a parameterization based on the rate of brittle fracture driven by viscous deformation and separately a scaling law for the kinetic energy of calving events. Overall, this study explores an important topic that has been the focus of considerable debate within the glaciology and ice sheet modeling community over the last several years and sheds light on the processes that might be driving the hypothesized ice cliff failure. A completed manuscript would likely be of interest to a broad community of ice sheet modelers and glaciologists.

We thank the reviewer for the consideration that they have given to our modeling and discussion of marine ice cliff instability. Addressing the comments that the reviewer provided has strengthened our results and clarified our key messages. We agree that the description of structural failure mechanisms leading to MICI will be of broad interest.

Major comments:

My major concern regards the validity of the parameterization determined at the end of the manuscript. Clearly a goal of this project is to derive a physically-based MICI parameterization (which has not been available previously, since the existing parameterization is based on few observations). However, to go from the simulations described in this study to a parameterization is a large leap that requires making a number of assumptions about how the simulations described here translate to large-scale viscous ice sheet models. In particular, the coupled ELMER-HiDEM simulations begin with an idealized square terminus geometry, which is then allowed to viscously deform, with periodic "checks" for fracture through simulations with the brittle version of HiDEM. Once the first full-thickness calving event happens, the simulation is stopped and the timing and size and the calving are converted into an "effective" calving rate ($C_{\hat{}}$ in table 1) which is fitted to determine the MICI parameterization. The critical assumption here is that the next calving event that occurs after this first simulated event will continue to follow this effective calving rate, even though it is starting from a very different terminus geometry and stress state (with the separate issue of the potential influence of the calved iceberg on subsequent stress state through a mélange-like buttressing effect). The authors have extrapolated from a single simulated calving event on time scales of hours to days to an effective calving rate that would be implemented in ice sheet models through the implicit assumption that many, many calving events (occurring over the time scale of a single viscous model time step of many weeks to months) would continue to occur with the frequency dictated by the initial idealized geometry. I just don't see how this is a physically meaningful or well-supported assumption. Perhaps the authors have a good explanation for why this is justified, but I don't find it in the current version of the manuscript. One possible way to justify this would be to do a long run of the coupled ELMER-HiDEM simulation, with many calving events and a continuously viscously deforming glacier geometry and show that all the calving events follow a calving rate that falls on the parameterized relationship defined

in this study. This would certainly be very convincing to showing that this is a robust parameterization that is not too dependent on the pristine initial conditions.

We thank the reviewer for thoroughly considering the calving rate law that we present. As we now note in our manuscript with greater emphasis and clarity, it will be essential for the calving rate law to be implemented in ice sheet models so that continuous viscous deformation and degrees-of-buoyancy are accounted for in the calving rate of any given outlet glacier. This is imperative because viscous deformation will evolve the glacier terminus after the initial calving event, as the reviewer notes. If great enough for a given ice thickness, viscous deformation may slow the rate of subsequent calving or put the terminus in a buoyant state (which we discuss further below). We state the conditions in which the calving rate law is to be applied in the Calving Rate Law and Implications for Glacier and Ice Sheet Retreat sections, and propose that the next step is for the ice sheet modeling community to decide how to best incorporate the law into models of Antarctic retreat.

In the Supplementary Information, we now present a series of test case simulations to demonstrate scenarios showing how viscous deformation affects subsequent calving events, following an initial event. These test cases include an example where surface lowering leads to a slower time to failure (TtF) for the subsequent calving event, a case where surface lowering through viscous deformation was great enough to put the calving face into a state of buoyancy, and a case of run-away MICI retreat following our derived calving rate law. This work strengthens the presented results and provides us further opportunity to explain the importance of considering viscous deformation when applying the law. It is documented in the revised manuscript and Supplementary Information.

Another issue I have arises from the "threshold" at which MICI starts (Line 90-92). How can we qualitatively distinguish between buoyancy-driven calving and the supposedly MICI-driven calving seen at $H_c > 136$ meters? Both involve full-thickness calving events. Why should one be included in the MICI parameterization and the other not? This makes the threshold of 136 meters seem somewhat arbitrary. Presumably this version of the parameterization would only make sense to implement in an ice sheet model that has a different parameterization for the calving that occurs below this threshold. How does that parameterization know when to stop calving?

The reviewer is correct. We suggest that two parameterizations for calving are incorporated in large-scale ice sheet models: one to represent buoyancy-driven calving and another to represent MICI retreat via structural ice cliff failure. Our parameterization is fit to retreat via the ice cliff failure observed in our simulation matrix. We do not consider buoyancy-driven failure in this parameterization. Buoyancy-driven failure can occur over a wide-range of situations and is not the result of a cliff failing due to stresses stemming from its own thickness. It is also unlikely to follow a similar relationship as we derive for MICI retreat rates with cliff height. For these reasons it is necessary to independently represent these two calving mechanisms in ice sheet models. Ice sheet modelers would need to ensure that there is a switch between the two parameterisations by considering degree-of-buoyancy. As we note above and in the manuscript, it will be important for ice sheet models to identify when, after an initial cliff failure event, the cliff height of a new calving face has been lowered through viscous deformation below the threshold at which MICI collapse will occur. A switch between the parameterizations would be necessary for these cases.

A lot of science is being compressed into a short manuscript, so I think the authors can work on the clarity of the manuscript and figures - more suggested edits in this direction can be found below.

Addressing the reviewers' comments on the manuscript text and figures has clarified our presented results as well as how they are intended to be used by the glacier and ice sheet modeling community. In addition, we have worked to 1) emphasize that our main contribution is the description of regimes through which ice

cliff failure, potentially leading to MICI retreat, will occur and 2) clarify language related to cliff failure vs MICI retreat or MICI collapse. The reviewer will notice that we now refer to variations of the term “structural ice cliff failure” when referring to a single calving event and MICI when referring to the longer-term retreat process.

Specific comments

Line 23: basins more than 1 km deep [since MICI wouldn't necessarily occur just in any overdeepened basin]

We have added text similar to that suggested by the reviewer.

Line 34: not sure what mounting means here

We have removed the term ‘mounting’.

Line 35: threshold being exceeded at a cliff height of...

We have incorporated the reviewer’s suggested edit.

Line 38: impossible is a strong word here, especially since you couple ELMER and HiDEM in a way that has the potential to be used for continental-scale simulations, as shown in Memon et al. 2019 including several of the co-authors on this study.

The term ‘impractical’ is now substituted for ‘impossible’.

Line 41-42: Not sure I follow the logic of this sentence

The sentence has been re-worded.

Line 63: delete comma

Deleted.

Line 100-101: Could you discuss more about how does this melange result relate to your other results? You've shown here that the likelihood of MICI-like calving is sensitively dependent on the stresses at the terminus. So, how will the glacier geometry and resulting effect on stress state (particularly when the terminus is not a perfect rectangle) change after the first calving event in your simulations? (This is related to my main point above).

In our idealised simulations, the stress state at the new calving face will be most impacted by changes to the surface slope caused by viscous deformation leading to the initial calving event. We have made a statement regarding how the magnitude of back force from mélange will need to increase to inhibit structural failure of these new calving faces.

Line 145: I’m a bit confused about why you use kinetic energy here, when you look at the actual magnitude of calving events in the previous simulations. I agree the kinetic energy is a useful metric, but why change between metrics halfway through the manuscript?

We changed metrics because it is not currently possible to determine TtF in simulations using HiDEM_{ve} due to the incongruent timescales of viscous and brittle failure processes. The kinetic energy metric allows

us to show that collapse acceleration similar to the viscous deformation – tensile failure mechanism (simulated in the Elmer/Ice – HiDEM_{be} workflow) is anticipated as terminus thicknesses increase. That being said, we have moved most of the discussion regarding kinetic energy as well as the associated figure to the Supplementary Information.

Line 167-174: this discussion of the viscous controls on T_{tf} would make a lot more sense in the previous section when you are actually talking about these simulations.

We understand that this discussion could also be moved to the Cliff Failure via Viscous Deformation and Tensile Failure section. We feel that it is appropriate to keep this text in the current section as it links the HiDEM_{ve} results with those reported earlier in the manuscript.

Line 174: Something else that is unclear to me here is that if the viscous mechanism dominates (but is fast) at cliff heights of 135-200 m, and then shear bands begin to dominate for H_c>200 m, would we expect that the shear bands would never occur in practice since the viscous mechanism should always occur first? I think I'm confusing myself about this, but this bears discussing I think.

The progression from viscous mechanisms to failure dominated by shear failure will depend generally on the balance between surface lowering through viscous deformation and the change in cliff height following a calving event. If a calving event causes the calving face to retreat to a location where the cliff height is taller than it was initially then it will be possible to move between the viscous deformation and shear band regimes of failure. This is now discussed in the shear localisation section.

Equation 2: I'm a bit confused about how you got this from table 1 (particularly the gray shaded cells), which shows that no MICI-like collapse occurs for H_c < 153 meters and T_{ice} - -20 or -10 C. Shouldn't I be set to zero at these heights?

We have decided not to add zeros for the heights at which ice cliff failure was not observed, as we think the current presentation most clearly indicates that structural ice cliff failure was not observed for these cases.

Line 235: Yes, you get higher wastage rates than DP16, but you also have a higher threshold for the onset of MICI calving and lower rates for cliffs below 200 meters (in Figure 2). Shouldn't this be pointed out?

It is now briefly noted that we find lower rates of retreat for these lower cliff heights.

Figure 1: There is a lot happening in this figure (could be split up), and the formatting is inconsistent between panels and backgrounds.

We have split Figure 1 into two figures. Figure 1 now only shows the viscous deformation – brittle failure mechanism along with demonstrating the Elmer/Ice – HiDEM_{be} modeling workflow. Figure 3 includes example model simulations in which shear band formation influenced cliff failure.

Figure 3c: Not sure what this panel is showing.

We have removed panel 3c as it was not illustrating anything that was not already adequately described in the text.

Figure 4: Not sure what the y axes are showing in this figure. Also the y-axis of panel b overlaps with panel a.

We have moved Figure 4 to the Supplementary Information. We have fixed the overlap of the two panels and direct the reader to the Methods for more information regarding the figures, including the y-axes.

Table 1: Why not also report the result of simulations for $H_c < 136$ meters? Does calving occur in any of these (even if it isn't MICI calving, see main comment above).

We have decided to report TtF, retreat magnitudes and calving rates associated with collapse via structural ice cliff failure only. We have not done a full set of simulations on buoyancy-driven calving below the temperature-dependent height thresholds at which MICI initiates. Comparing retreat via buoyancy-driven calving and MICI has potential to serve as the base of a potential future study.

Reviewer 2:

General comments

Marine Ice Cliff Instability (MICI) is currently a hot topic in Antarctic glaciology. In two papers (Pollard et al, 2015 and DeConto and Pollard, 2016) a parameterisation of this mechanism was introduced, based on earlier work of one of the co-authors of this manuscript (Bassis and Walker, 2011). MICI was introduced to enable to simulate Pliocene high sea-level stands that can only be explained by large mass losses from the Antarctic ice sheet. The introduction of MICI in DP16 (DeConto and Pollard, 2016) has profoundly shaken the modelling community, as the mechanism potentially results in future sea-level rise estimates of almost an order of magnitude larger compared with other studies. Therefore, a better comprehension of this mechanism is urgently needed, which this manuscript by Crawford et al. tries to do

This novel study investigates MICI (Marine Ice Cliff Instability) collapse for brittle and viscous processes (both separated and mixed) using two different ice flow models, i.e. Elmer/Ice for viscous deformation, and HiDEM for brittle deformation. For undamaged ice, viscous deformation is needed to initiate cliff collapse (by forming a fore bulge), but viscous deformation also imposes tensile stress on the glacier surface, which initiates crevassing (and damages the ice). The HiDEM model is used for two purposes: (i) to investigate cliff collapse for undamaged ice (HiDEM_{be}) and for damaged ice through shear-band deformation (HiDEM_{ve}). Based on the result from the undamaged ice experiments, the authors manage to establish a new calving law based on brittle and viscous deformation, which is compared to calving laws for MICI as parameterised by DP16. They furthermore look at the back force of ice mélange that may lead to calving face stability.

The paper is important for providing the reader with new derived calving laws based on MICI, but their use are limited to the undamaged case. These calving rates are also less important than those used in DP16 and cliff failure generally occurs for higher cliffs than the range in DP16. It is a pity that effects of shear failure cannot be accounted for in these laws, as for damaged ice the authors limit this effect by representing kinetic energy as a function of cliff height. In the discussion, the authors make the point that shear band formation may well increase the calving rate (comparable to DP16, but not quantified other than through kinetic energy) and that on the other hand ice mélange formed by the calved ice may exert sufficient large stresses to stabilise the cliffs again.

The reviewer has provided an excellent summary of our presented work and its place within the larger conversation regarding marine ice cliff instability and how it could dramatically reshape Antarctica. We greatly appreciate the reviewer's attention to our manuscript and we have worked to address the reviewer's thoughtful comments.

We have adjusted the structure of the manuscript to discuss shear failure in HiDEM in more detail. Developments to the HiDEM model allow us to now simulate shear failure in HiDEM_{be}. However, it is not

possible at this time to validate the parameters that are tuned to allow shear failure to be observed in our model runs. Therefore, we use these results to add to the discussion of the various modes through which MICI could transpire and comment on how shear failure will likely impact retreat rates. We now present our calving law at the end of the manuscript. The calving law is, as before, derived from simulations of undamaged ice calving through tensile failure. We are comfortable providing this to the ice sheet modeling community as the equation is derived from a modeling workflow that allows us to explicitly determine the time to failure and represents the most conservative structural failure mechanism of the modes discussed in our paper.

Major comments

Based on the HIDEM_{ve} experiments, a further quantification beyond the energy scaling, i.e., providing cliff retreat rates as a function of cliff height, cannot be 'properly' done. If it cannot be done properly, could an estimate or range be given? Because this would underpin further statements on the comparison with parameterisations of DP16.

We do not feel that it would be good practice to provide an estimate of retreat rate associated with shear failure at this time. We hope that future work will be able to provide such an estimate.

The same accounts for the effect of ice mélange. In Fig. 3 an example of one of the experiments is shown, but how does this summarise for all experiments?

Table 1 provides the back force values necessary to halt ice cliff failure for a range of cliff heights. A description of how the required back force changes with cliff height is provided in the Cliff Failure via Viscous Deformation and Tensile Failure.

The paper is quite technical in nature and could benefit from more context. For instance, why is MICI important and why it is controversial; reference to Edwards et al (2019) re: difficulties in reconciling with other SLR estimates; why MICI has been introduced for explaining Pliocene SLR, ... That context is necessary for a publication in a journal targeting a wider public.

We agree with the reviewer that additional context is helpful for the broad audience that this journal attracts. This context is now included in the introduction.

The paper also mixes up tenses. Most of it is written in the past tense, but the discussion is in the present tense. For clarity and ease of flow it would be preferential to write everything in present tense.

We thank the reviewer for pointing this out. We now use the present tense throughout the manuscript.

Detailed remarks

Line 24: I don't know any other ice sheet model than Pollard et al. (2015) and DP16 who have included MICI in simulations. In my mind, only one model uses simple parameterisations, and the other models don't include MICI.

Line 34: explain 'mounting glaciostatic stress'

We have removed the term 'mounting' as both reviewers have highlighted that this term is not standard.

Line 64 (and others): surface-to-base velocity ratios instead of surface : base

Corrected here and elsewhere.

Line 174: reached.

Corrected, thank you.

Line 178: remove period after 'equation'.

Corrected, thank you.

Lines 191-193: It should be added that the ice mélange could have the opposite effect to put the 'conservative' measure in perspective.

This has been addressed in the concerned section, which now includes more discussion about the influence of mélange and the need to represent mélange in ice sheet models.

Line 213: the opening sentence is weird. I wouldn't even call it 'new', as it may well have occurred in the past (e.g., Pliocene).

This sentence has been changed and now simply reads, "Climate change is transforming the Antarctic continent."

Line 214: 'precarious situation' is not a scientific term to describe what the effect of ice shelves can be. To give a bit more context here, a recent paper by Sun et al. (2020; Journal of Glaciology) has demonstrated in a multi-model context that loss of ice shelves due to MISI (and without MICI) leads to multi-meter SLR on short time scales (hundreds of years). One could therefore argue that key in the stability of WAIS is the point at which ice shelves collapse due to atmospheric/oceanic forcing, more so than the fact whether MICI kicks in or not.

We agree that this wording was not appropriate and have modified this sentence.

Line 365: It should be stated why a 4 by 3 km domain with passive sidewalls is representative for Thwaites glacier that is much wider than the domain used.

An explanation as to why this domain size and passive sidewalls were used in these simulations is now included. The domain size was restricted by computational power.

Line 371: C is a friction coefficient. Furthermore, a linear bed friction law is used, while several studies show that basal friction is probably more plastic. What would be the effect of nonlinear sliding on calving rates or do they not have an effect?

Friction is implicated in calving rates and we expect that changing the calving law could modify our quantitative results though, qualitatively, the relationships that we describe would remain true. We have added a note that future studies could assess the impact of different sliding laws on retreat rates.

Line 481: cases.

Typo corrected.

REVIEWER COMMENTS

Reviewer #1 (Remarks to the Author):

This revised manuscript uses a state-of-the-art model of brittle fracture of glaciers, coupled with a model of viscous glacier deformation to simulate calving at tidewater glacier termini. The result is a parameterization of calving rate based on the first calving event time-to-fracture in the coupled viscous-brittle framework.

The authors have made many valuable changes to the manuscript that include streamlining it, clarifying certain aspects of the technical description, making the figures more readable, and conducting more simulations in response to my previous review. In this review I will focus on two major lingering issues that haven't been entirely resolved by the impressive revisions.

1. My previous review had expressed concern that the parameterization was based on a small set of single calving events from a highly idealized initial condition. The question I raised was what would happen after the first calving event as the geometry and stress of the terminus continued to evolve? Would the more realistic cliff geometry produced after the first calving event follow the calving rate law suggested by the idealized geometries? The authors have extended three of their simulations in the way I requested to answer these questions. I have two takeaways from these simulations:

(a) The only extended simulation for which the authors report the time-to-fracture and calving event size for the second calving event is the second one ($H=2000$ m, $T = -20$ C). This second calving produces an implied calving rate of 13 m/day, where the parameterization based on first calving events predicts 33 m/day (lines 59-63 of the supplement). This is a factor of 2.5 difference between the prediction from the parameterization and the rate implied by the simulated second event. That seems like a large error! I get that on the log-scale of the power-law parameterization, it is "close", but it implies that the changing geometry and stress at the terminus (beyond just the subaerial cliff height) cause significant $O(1)$ changes in the rate and extent of structural cliff failure beyond what would be predicted by simulations from a highly idealized state. If anything, I think this shows that extrapolating a calving law based on 4-10 idealized calving events may be prone to large errors/uncertainties. The model of the authors at least allows those uncertainties to be estimated by looking at many calving events after the first calving event. Could this be done in a more rigorous manner to even begin to estimate the scatter/uncertainty in the parameterization? At the very least, the magnitude of this large uncertainty needs to be acknowledged in the main text by the authors so that modelers who want to use this parameterization have an idea of the magnitude of uncertainty.

It is also notable that this one data point produces calving rates that are considerably lower than the parameterization prediction, implying that the tendency of viscous deformation to modify the cliff stress and geometry reduces the propensity for calving. This cuts against the argument of the authors that the parameterization represents a lower bound, rather it may be biased high by the artificially unstable geometry of the initial condition. This issue should also be addressed.

(b) The fact that two of extended simulations ($H = 1000$ m $T = -5$ C and $H = 2000$ m $T = -20$ C) lead to a cliff height that is lower for each subsequent calving event is notable because it implies that for these initial cliff heights, MICI is not an instability at all. That is to say, even though "structural failure" occurs at these heights, this structural failure leads to lower cliff heights, which is, in effect, a negative feedback preventing runaway calving. As in the $H = 1000$ m case, eventually (potentially within just a few calving events over a few weeks or months, i.e. a few time steps in a large ice sheet model) this will lead to cliff heights low enough to prevent further structural failure. The third extended simulation does show that runaway is possible, but it implies that the cliff height threshold for runaway is much higher than previously argued by D&P (and requires cold ice). This is a very different prediction than the MICI as presented in the two DeConto and Pollard papers on the subject, and noting this difference would probably be of interest to the community.

2. I think that the authors have done a better job in this version of the manuscript explaining that they are putting forward this new calving parameterization as separate from a buoyant calving parameterization in large scale ice sheet models. What is still unclear to me is, in the HiDEM simulations, how do the authors decide what is "buoyant" calving and what is "structural failure"? Specifically, on lines 115-117 it is stated "The retreat of simulated glaciers thinner than this threshold ($H_c < 127$) is dominated by buoyancy-driven calving". In the simulation output, how does one distinguish a "buoyant calving event" from structural failure? One could argue that all calving represents structural failure (I get that this is a bit specious, given the specific argument about shear failure in Bassis & Walker 2012), so can the authors clarify for those who are not intimately familiar with the literature how they draw this distinction, which is critical to setting the threshold at which this parameterization "kicks in"?

I think it would be most productive to address these conceptual issues in the text before proceeding with further minor edits.

Reviewer #2 (Remarks to the Author):

I would like to thank the authors for the improvements they made to the manuscript and the care they took in incorporating the various comments from the reviewers. Overall, the manuscript also is clearer and reads better and I therefore recommend the paper for publication. I only have a couple of minor remarks and typos to report.

The use of B: In Table 1 and the main text you use B_m , B_l and B_h as nominal values for the surface-to-base velocity ratios. However, in Equation 1 and Figure 1, B_f , B_n and B_h appear, apparently for describing the same set. What makes it more confusing is that in Figure 1, B is defined as : B_n , B_f and B_h represent low-slip (normal), approaching frozen bed and high-slip bed condition. Wouldn't it be better to define it as normal slip, approaching frozen and high slip, respectively? The low-slip (normal) condition seems a bit confusing, as for me approaching frozen is equivalent to low slip. On lines 104-106 (and methods lines 537-545) you use mid range B_m , great basal slip B_l (low velocity ratio) and less basal slip B_h (high velocity ratio), which is contradicting with the B_h of the figure which represents the opposite. Either you present the B values as a slip condition or as a surface-to-base condition, but mixing them up makes no sense with the subscripts. Finally, in the Methods section (line 616) B_N , B_F and B_H appear, with capital subscripts. Some cleaning is in order.

Line 48: continental-scale ice-sheet models

Line 63: high, calving

Line 65: It is (capital)

Line 119: either 'decreasing with increasing H_c ' or 'show that the T_tF decreases with ...'.

Line 121: you removed 'mounting' in the introduction, but here it still appears.

Reply to Reviewers: Manuscript NCOMMS-20-35781A by Crawford, Benn, Todd, Åström, Bassis and Zwinger: “Marine ice-cliff instability modeling shows mixed-mode ice-cliff failure and yields calving rate law”

Original reviewer comments are in black.

Author replies are in blue.

Reviewer 1:

General comments:

This revised manuscript uses a state-of-the-art model of brittle fracture of glaciers, coupled with a model of viscous glacier deformation to simulate calving at tidewater glacier termini. The result is a parameterization of calving rate based on the first calving event time-to-fracture in the coupled viscous-brittle framework.

The authors have made many valuable changes to the manuscript that include streamlining it, clarifying certain aspects of the technical description, making the figures more readable, and conducting more simulations in response to my previous review. In this review I will focus on two major lingering issues that haven't been entirely resolved by the impressive revisions.

We would like to thank the reviewer for thoroughly engaging with our work. We agree that the previous changes, which addressed the reviewer's previous comments, were valuable and improved the quality of the article. Our responses to the current round of comments are included in-line below.

Major comments:

1. My previous review had expressed concern that the parameterization was based on a small set of single calving events from a highly idealized initial condition. The question I raised was what would happen after the first calving event as the geometry and stress of the terminus continued to evolve? Would the more realistic cliff geometry produced after the first calving event follow the calving rate law suggested by the idealized geometries? The authors have extended three of their simulations in the way I requested to answer these questions. I have two takeaways from these simulations:

(a) The only extended simulation for which the authors report the time-to-fracture and calving event size for the second calving event is the second one ($H=2000$ m, $T = -20$ C). This second calving produces an implied calving rate of 13 m/day, where the parameterization based on first calving events predicts 33 m/day (lines 59-63 of the supplement). This is a factor of 2.5 difference between the prediction from the parameterization and the rate implied by the simulated second event. That seems like a large error! I get that on the log-scale of the power-law parameterization, it is "close", but it implies that the changing geometry and stress at the terminus (beyond just the subaerial cliff height) cause significant $O(1)$ changes in the rate and extent of structural cliff failure beyond what would be predicted by simulations from a highly idealized state. If anything, I think this shows that extrapolating a calving law based on 4-10 idealized calving events may be prone to large errors/uncertainties. The model of the authors at least allows those uncertainties to be estimated by looking at many calving events after the first calving event. Could this be done in a more rigorous manner to even begin to estimate the scatter/uncertainty in the parameterization? At the very least, the magnitude of this large uncertainty needs to be acknowledged in the main text by the authors so that modelers who want to use this parameterization have an idea of the magnitude of uncertainty.

It is also notable that this one data point produces calving rates that are considerably lower than the parameterization prediction, implying that the tendency of viscous deformation to modify the cliff stress and geometry reduces the propensity for calving. This cuts against the argument of the authors that the parameterization represents a lower bound, rather it may be biased high by the artificially unstable geometry of the initial condition. This issue should also be addressed.

We agree with the reviewer that it is important to report uncertainty wherever possible. We would like to note that our presented calving retreat rate law has its own residual error, as it is a smooth power-law fit to a scattered set of observations. Derivation from the fitted model is therefore expected for individual calving events that occur to initial glacier termini as well as the newly-exposed cliffs of evolved glacier geometries.

A plot of the residuals associated with the fitted retreat rate law is now included. The residual plot shows a high degree of fit for lower cliff heights and that the fit decreases with increasing cliff height. We have corrected the value of the predicted retreat rate for the $T_{ice} = -5$ °C scenario and have added discussion in the main text and Supplementary Information regarding the model fit, an outlier data point, and error in predicted retreat rates. The latter includes discussion of the secondary cliff failure events. The calving retreat rates associated with the two simulations of secondary cliff failure events are not consistently over- or under-predicted by the fitted retreat rate law. The errors associated with these individual, secondary events are also of similar magnitude to the residual error associated with the model fit to the initial calving retreat rates.

As we now point out in the text, the model error is large for some simulations on an individual basis. The temporal resolution of our Elmer/Ice simulations influences the large absolute errors associated with the simulations of the greatest tested cliff heights. We put these larger variances for the greater cliff heights into context by emphasizing a key point: the trend of our simulations shows that the retreat of these extraordinary cliff heights will be alarming rapid if viscous deformation and/or mélange buttressing do not counteract cliff instability. Changes to surface slope and cliff height resulting from viscous deformation will modify the cliff stress and geometry, as the reviewer states, and will be crucial to consider as the processes may slow or stabilise these initially unstable termini. This continual evolution may mean that the extreme cliff heights associated with these rapid retreat rates are not reached.

Importantly, the calving retreat rates associated with the secondary cliff-failure events correctly follow their respective scenarios' trendline. We stress that the fitted law captures the trends in retreat rate with increasing thickness, warming ice temperature and altered basal slip. As mentioned in Pollard et al. (2015), we do not expect ice-sheet modelers to be interested in individual events but rather concerned that the overall trend of retreat with cliff height is adequately represented. We still consider the law to be conservative, as it did not consistently over-predict the retreat rates of secondary cliff failure events. Furthermore, the law does not reflect the impact of shear localisation, which will accelerate ice-cliff failure.

We now point out in the main text that the law can be refined in the future. We greatly thank the reviewer for pushing us to consider these topics. The paper is strengthened greatly by addressing their comment.

(b) The fact that two of the extended simulations ($H = 1000$ m $T = -5$ C and $H = 2000$ m $T = -20$ C) lead to a cliff height that is lower for each subsequent calving event is notable because it implies that for these initial cliff heights, MICI is not an instability at all. That is to say, even though "structural failure" occurs at these heights, this structural failure leads to lower cliff heights, which is, in effect, a negative feedback preventing runaway calving. As in the $H = 1000$ m case, eventually (potentially within just a few calving events over a few weeks or months, i.e. a few time steps in a large ice sheet model) this will lead to cliff heights low enough to prevent further structural failure. The third extended simulation does show that runaway is possible, but it implies that the cliff height threshold for runaway is much higher than previously

argued by D&P (and requires cold ice). This is a very different prediction than the MICI as presented in the two DeConto and Pollard papers on the subject, and noting this difference would probably be of interest to the community.

The reviewer brings up a key point that we want to impart with readers: that the occurrence of a single structural ice-cliff failure event does not equate to initiation of the self-sustaining retreat process known as marine ice-cliff instability (MICI). We now reiterate our definitions of structural ice-cliff failure and MICI in the Cliff Failure via Viscous Deformation and Tensile Failure section and refer throughout the article to “structural ice-cliff failure” and “run-away MICI retreat” (or similar) to highlight this distinction. The point that viscous deformation can stabilise newly-exposed ice cliffs or slow their failure is now repeated as well.

We note that Pollard et al. (2015) and DeConto and Pollard (2016) routinely refer to “cliff-failure” and “structural ice-cliff failure”, though the term “structural ice-cliff failure” may be seen to be used somewhat interchangeably with “MICI” in DeConto and Pollard (2016). We had also, unhelpfully, continued this confusion in the conclusion of our article. This has been edited to clarify that we are referring to their ice-cliff failure parameterization.

In terms of model simulations, we believe that we are considering structural ice-cliff failure and MICI in the same terms as Pollard et al (2015) and DeConto and Pollard (2016). Our law and the DeConto and Pollard parameterization both consider the rate of retreat given an exposed ice-cliff height and expect MICI initiation once retreat is directed into over-deepening basins. Our work focuses on the failure and evolution of ice cliffs near the height threshold at which they are susceptible to structural failure, which is not a focus of the earlier studies mentioned here.

We have provided only one example scenario in which run-away MICI retreat initiates and we do not want to state that such runaway retreat will only initiation for such a situation ($T_{ice} = -20$ °C). Instead, we bring text from the Supplementary Information to the main text to make the point that the magnitude of influential surface lowering will be dictated by numerous factors that impact dynamic thinning.

In summary, we expect that acceleration and de-acceleration of ice-cliff failure, especially around the cliff-height threshold, would be seen in ice-sheet models that simulate dynamic thinning at an appropriate time-step when incorporating either our retreat rate law or the DeConto and Pollard (2016) parameterization. It will be important to consider the timestep at which ice-sheet models are run, as this will influence the degree of terminus evolution that occurs between inspection for structural ice-cliff failure conditions.

2. I think that the authors have done a better job in this version of the manuscript explaining that they are putting forward this new calving parameterization as separate from a buoyant calving parameterization in large scale ice sheet models. What is still unclear to me is, in the HiDEM simulations, how do the authors decide what is "buoyant" calving and what is "structural failure"? Specifically, on lines 115-117 it is stated "The retreat of simulated glaciers thinner than this threshold ($H_c < 127$) is dominated by buoyancy-driven calving". In the simulation output, how does one distinguish a "buoyant calving event" from structural failure? One could argue that all calving represents structural failure (I get that this is a bit specious, given the specific argument about shear failure in Bassis & Walker 2012), so can the authors clarify for those who are not intimately familiar with the literature how they draw this distinction, which is critical to setting the threshold at which this parameterization "kicks in"?

We agree that it is worthwhile to elaborate on the distinction between buoyancy-driven calving and structural ice-cliff failure. These distinctions are now clarified in the Cliff Failure via Viscous Deformation and Tensile Failure section as well as in the Supplementary Information.

I think it would be most productive to address these conceptual issues in the text before proceeding with further minor edits.

Reviewer 2:

General comments

I would like to thank the authors for the improvements they made to the manuscript and the care they took in incorporating the various comments from the reviewers. Overall, the manuscript also is clearer and reads better and I therefore recommend the paper for publication. I only have a couple of minor remarks and typos to report.

We thank the reviewer for their positive response and the time and effort that they invested to their critique of our article. Addressing both rounds of the reviewer's comments has substantially improved the work.

Minor comments

The use of B: In Table 1 and the main text you use B_m , B_l and B_h as nominal values for the surface-to-base velocity ratios. However, in Equation 1 and Figure 1, B_f , B_n and B_h appear, apparently for describing the same set. What makes it more confusing is that in Figure 1, B is defined as : B_n , B_f and B_h represent low-slip (normal), approaching frozen bed and high-slip bed condition. Wouldn't it be better to define it as normal slip, approaching frozen and high slip, respectively? The low-slip (normal) condition seems a bit confusing, as for me approaching frozen is equivalent to low slip. On lines 104-106 (and methods lines 537-545) you use mid range B_m , great basal slip B_l (low velocity ratio) and less basal slip B_h (high velocity ratio), which is contradicting with the B_h of the figure which represents the opposite. Either you present the B values as a slip condition or as a surface-to-base condition, but mixing them up makes no sense with the subscripts. Finally, in the Methods section (line 616) B_N , B_F and B_H appear, with capital subscripts. Some cleaning is in order.

We thank the reviewer for pointing out inconsistencies in how we referred to basal slip conditions. As suggested, we now consistently use B_n , B_f and B_h to represent normal slip, approaching frozen, and high slip basal conditions. Table 1, the caption of Figure 2 and the Methods section have been updated and conditions are no longer defined by the surface : base velocity ratios. We now mention the surface : bed velocity ratios to describe the impact of increasing or decreasing basal slip on overhang development.

Line 48: continental-scale ice-sheet models

Corrected, thank you.

Line 63: high, calving

Corrected, thank you.

Line 65: It is (capital)

We do not believe that a correction is necessary with the capitalisation.

Line 119: either 'decreasing with increasing H_c ' or 'show that the TtF decreases with ...'.

Corrected, thank you. The sentence now uses, 'decreasing with increasing H_c '...

Line 121: you removed 'mounting' in the introduction, but here it still appears.

The term 'mounting' has been removed.

REVIEWERS' COMMENTS

Reviewer #1 (Remarks to the Author):

This manuscript on "Marine ice-cliff instability modeling shows mixed-mode ice-cliff failure and yields calving rate law" is a revised version of an earlier manuscript. It has addressed all of the substantive issue raised in previous reviews, and as a result it is nearly ready for publication. I have a few minor comments on the text and language used throughout the manuscript. However, I leave it to the discretion of the editor to decide the extent to which these changes need to occur and leave them to review subsequent versions of the manuscript.

One issue I have with the language throughout the manuscript and in the title is the use of the term "rate law". While it has become common practice to call parameterisations "laws" in glaciology (particular when it comes to flow or calving), I would not describe the functional relationship derived in this study a "law". For one, it is completely based on idealized simulations. A "law" is typically a function which describes all available data and which is derived from observations of a particular process. On the other hand, the use of a high-fidelity numerical model to constrain the functional relationship between two variables (in this case time-averaged retreat rate and ice cliff height) for use in lower fidelity models is described in almost every other field (including the rest of Earth sciences) as a "parameterisation". My suggestion would be to change the use of the term "rate law" throughout the manuscript to "parameterisation".

Minor issues by line number:

Line 26: here and elsewhere you use the word "conservative" before explaining what you mean. This is confusing because conservative is also often use to mean some equation in which a quantity does not change over time. I think it would just make more sense to remove it here and wait until later in the manuscript to explain what you mean.

Line 36: the use of instability to explain ice cliff failure is confusing here (particularly because you are trying to distinguish cliff failure from the marine ice cliff instability). I think "fracture" would make more sense.

Line 39: the modern observational record

Line 41: when ice-cliff failure is included with [as you say, this parameterization isn't of MICI, but of cliff failure]

Line 54: will break if strain, bending...

Line 71: same issue above with conservative

Line 94: basal friction conditions

Line 112: are used to calculate time-averaged retreat rate

Line 128: can you explain here in the main text in one sentence how melange is incorporated? (i.e. how is buttressing force prescribed in HiDEM)

Line 129: scientific notation superscripts are confusing (particular with journal citation formatting)

188: "domain increases in thicknesses" - this is confusing phrasing

Line 210: based on the minimum height of ice-cliff failure found through...

Line 240: conjunction with model representation of ice surface lowering via dynamic thinning at a sufficiently high temporal resolution, which...

Line 247: again with use of conservative here (its even more confusing here)

Line 247: delete "tendential"

Line 250: delete "lastly" - this isn't the last point made in this paragraph

Line 259: should probably cite some of the papers which have recently attempted to simulate melange in ice sheet models (Pollard et al. in GMD, Amundson and Burton in JGR)

Line 265: expose tall ice cliffs

Line 270: leading to iceberg calving that is distinct from buoyant calving (CITE)

Line 275: release and terminus retreat incorporated

Line 277: states of rapid failure with increasing

Line 278: In all cases, ice cliff failure produces copious melange

Line 281: evacuation is a rate-limiting processes

Line 293-294: I'm confused by the point of this sentence because the "rate law" is not defined below $H_c=135$, and you say that cliff failure does not occur at cliff heights of 80-100 meters in these simulations. Not sure this sentence is accurate (or necessary).

Figure 2: the residual inset has the wrong units on y-axis

Reply to Reviewers: Manuscript NCOMMS-20-35781A by Crawford, Benn, Todd, Åström, Bassis and Zwinger: “Marine ice-cliff instability modeling shows mixed-mode ice-cliff failure and yields calving rate relationship”

Original reviewer comments are in black.

Author replies are in blue.

Reviewer 1:

General Comments

This manuscript on "Marine ice-cliff instability modeling shows mixed-mode ice-cliff failure and yields calving rate law" is a revised version of an earlier manuscript. It has addressed all of the substantive issue raised in previous reviews, and as a result it is nearly ready for publication. I have a few minor comments on the text and language used throughout the manuscript. However, I leave it to the discretion of the editor to decide the extent to which these changes need to occur and leave them to review subsequent versions of the manuscript.

We thank the reviewer for the great consideration that they have given to our work. The article has been improved through addressing the reviewer's comments.

One issue I have with the language throughout the manuscript and in the title is the use of the term "rate law". While it has become common practice to call parameterisations "laws" in glaciology (particular when it comes to flow or calving), I would not describe the functional relationship derived in this study a "law". For one, it is completely based on idealized simulations. A "law" is typically a function which describes all available data and which is derived from observations of a particular process. On the other hand, the use of a high-fidelity numerical model to constrain the functional relationship between two variables (in this case time-averaged retreat rate and ice cliff height) for use in lower fidelity models is described in almost every other field (including the rest of Earth sciences) as a "parameterisation". My suggestion would be to change the use of the term "rate law" throughout the manuscript to "parameterisation".

The reviewer is correct in that the terms “law” and “parameterization” seem to be used interchangeably in the field of glaciology when equations are fit to observational or modelled data sets. We have taken the reviewer's suggestion and now refer to our calving retreat rate parameterization throughout our work.

Minor issues by line number:

Line 26: here and elsewhere you use the word "conservative" before explaining what you mean. This is confusing because conservative is also often use to mean some equation in which a quantity does not change over time. I think it would just make more sense to remove it here and wait until later in the manuscript to explain what you mean.

We have removed the word “conservative” in this location.

Line 36: the use of instability to explain ice cliff failure is confusing here (particularly because you are trying to distinguish cliff failure from the marine ice cliff instability). I think "fracture" would make more sense.

We understand how the use of "instability" multiple times in this sentence can be confusing. The term "stress imbalance" now replaces the concerned instance of "instability".

Line 39: the modern observational record

The word "modern" has been added.

Line 41: when ice-cliff failure is included with [as you say, this parameterization isn't of MICI, but of cliff failure]

This substitution has been made.

Line 54: will break if strain, bending...

The sentence segment now reads as the reviewer suggests.

Line 71: same issue above with conservative

The word "conservative" has been removed here.

Line 94: basal friction conditions

The word "friction" has been added.

Line 112: are used to calculate time-averaged retreat rate

The term "time-averaged" has been added.

Line 128: can you explain here in the main text in one sentence how melange is incorporated? (i.e. how is buttressing force prescribed in HiDEM)

We have added two sentences that explain how mélange is incorporated in our idealised simulations.

Line 129: scientific notation superscripts are confusing (particular with journal citation formatting)

We feel that this is an appropriate format for reporting our results and that the context in which the superscripts are used is apparent.

Line 188: "domain increases in thicknesses" - this is confusing phrasing

We now include the term "simulated glacier domain..." to remove confusion.

Line 210: based on the minimum height of ice-cliff failure found through...

We have incorporated the following phrasing, "based on the minimum ice-cliff height at which failure occurs, which is found through...".

Line 240: conjunction with model representation of ice surface lowering via dynamic thinning at a sufficiently high temporal resolution, which...

This edit has been made.

Line 247: again with use of conservative here (its even more confusing here)

The use of the word conservative is necessitated at this location, as the sentence explains additional factors of our work that result in the derived calving rate relationship being conservative in nature.

Line 247: delete "tendential"

Deleted.

Line 250: delete "lastly" - this isn't the last point made in this paragraph

Deleted.

Line 259: should probably cite some of the papers which have recently attempted to simulate melange in ice sheet models (Pollard et al. in GMD, Amundson and Burton in JGR).

These references have been added.

Line 265: expose tall ice cliffs

This now reads, "expose tall and unstable ice cliffs".

Line 270: leading to iceberg calving that is distinct from buoyant calving (CITE)

This phrasing has been incorporated and we now cite Benn et al. (2017) and Wagner et al. (2015).

Line 275: release and terminus retreat incorporated

The phrase now reads, "kinetic energy release and terminus retreat as ice thickness increases."

Line 277: states of rapid failure with increasing

The replacement of "instability" with "failure" has been made.

Line 278: In all cases, ice cliff failure produces copious melange

The replacement of "instability" with "ice-cliff failure" has been made.

Line 281: evacuation is a rate-limiting processes

The replacement of "the" with "a" has been made.

Line 293-294: I'm confused by the point of this sentence because the "rate law" is not defined below $H_c=135$, and you say that cliff failure does not occur at cliff heights of 80-100 meters in these simulations. Not sure this sentence is accurate (or necessary).

We now provide some further details in this sentence to clarify that we are referring to cliff heights between 135 (the minimum height at which we observed cliff failure in our simulations) and 200 m.

Figure 2: the residual inset has the wrong units on y-axis

Corrected, thank you.